# MapEval: A Map-Based Evaluation of Geo-Spatial Reasoning in Foundation Models

**Mahir Labib Dihan**[1]  **Md Tanvir Hassan**[1][*]  **Md Tanvir Parvez**[2][*]  **Md Hasebul Hasan**[1]  **Md Almash Alam**[3]
**Muhammad Aamir Cheema**[4]  **Mohammed Eunus Ali**[1][4]  **Md Rizwan Parvez**[5]

## Abstract

Recent advancements in foundation models have improved autonomous tool usage and reasoning, but their capabilities in map-based reasoning remain underexplored. To address this, we introduce **MapEval**, a benchmark designed to assess foundation models across three distinct tasks—textual, API-based, and visual reasoning—through 700 multiple-choice questions spanning 180 cities and 54 countries, covering spatial relationships, navigation, travel planning, and real-world map interactions. Unlike prior benchmarks that focus on simple location queries, MapEval requires models to handle long-context reasoning, API interactions and visual map analysis, making it the most comprehensive evaluation framework for geospatial AI. On evaluation of 30 foundation models, including Claude-3.5-Sonnet, GPT-4o, Gemini-1.5-Pro, none surpasses 67% accuracy, with open-source models performing significantly worse and all models lagging over 20% behind human performance. These results expose critical gaps in spatial inference, as models struggle with distances, directions, route planning, and place-specific reasoning, highlighting the need for better geospatial AI to bridge the gap between foundation models and real-world navigation. All the resources are available on the project website.

---

[*]Equal contribution  [1]Department of Computer Science and Engineering Bangladesh University of Engineering and Technology (BUET) [2]Statistics, Islamic University, Bangladesh [3]Bangladesh Computer Council (BCC) [4]Faculty of Information Technology, Monash University, Melbourne, Australia [5]Qatar Computing Research Institute (QCRI). Correspondence to: Mahir Labib Dihan <mahirlabibdihan@gmail.com>, Mohammed Eunus Ali <mohammed.eunus.ali@gmail.com>, Md Rizwan Parvez <mparvez@hbku.edu.qa>.

*Proceedings of the 42nd International Conference on Machine Learning*, Vancouver, Canada. PMLR 267, 2025. Copyright 2025 by the author(s).

## 1. Introduction

Recent advancements in foundation models, particularly large language models (LLMs) and vision-language models (VLMs), are significantly enhancing the capabilities of AI systems in autonomous tool usage (Qin et al.; Yao et al., 2023) and reasoning (Lu et al.; Wei et al., 2022). These developments facilitate the automation of everyday tasks through natural language instructions, especially in domains that require interaction with specialized tools like map services.

As platforms such as Google Maps or Apple Maps have become ubiquitous for accessing various location-based services (a.k.a tools/APIs) —ranging from finding nearby restaurants to determining the fastest routes between origins and destinations—there has been a growing interest in integrating maps with foundation models (Xie et al.; Zheng et al., 2024). A couple of recent initiatives, such as WebArena (Zhou et al.) and VisualWebArena (Koh et al., 2024), have introduced new tasks that involve map usage in practical scenarios.

However, despite the widespread adoption of map services and the promising potential of interactions between foundation models (e.g., LLMs and VLMs) and these services, no studies have rigorously tested the capabilities of foundation models in location or geo-spatial reasoning. This gap is critical, as effective map-based reasoning can optimize navigation, facilitate resource discovery, and streamline logistics in everyday life. Addressing this gap is essential for advancing the practical utility of AI in real-world applications.

We introduce MapEval, a novel benchmark designed to evaluate the geo-spatial reasoning capabilities of foundation models and AI agents in complex map-based scenarios. MapEval addresses a critical gap in existing benchmarks by evaluating models' ability to process heterogeneous geospatial contexts, perform compositional reasoning, and interact with real-world map tools. It features three task types—API, Visual, and Textual—that require models to collect world information via map tools, a deep visual understanding, and reason over diverse geo-spatial data (e.g., named entities, coordinates, operational hours, distances, routes, user reviews/ratings, map images), all of which remain chal-

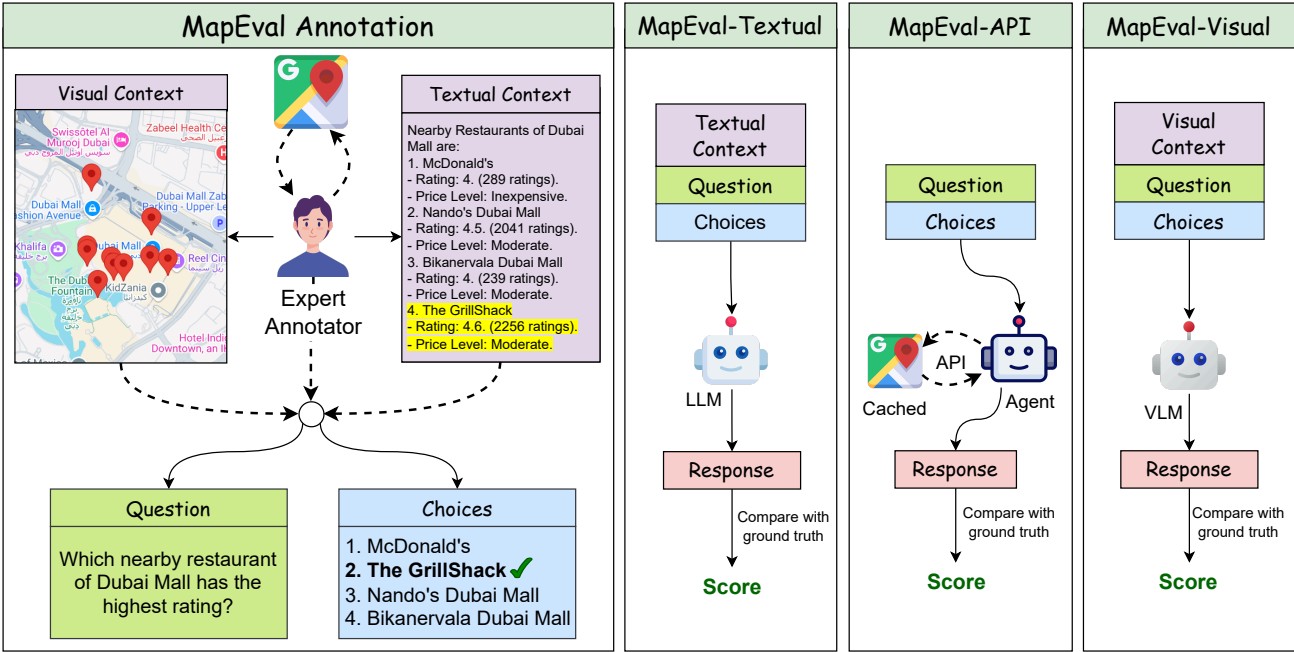

*Figure 1.* Overview of MapEval. On the left, we show the annotation process, where an expert gathers either visual snapshots or textual data from Google Maps to create multiple-choice questions with ground truth labels. On the right, we depict the evaluation process and input/output for the three benchmark tasks in MapEval.

lenging for state-of-the-art foundation models. Comprising 700 unique multiple-choice questions across 180 cities and 54 countries, MapEval reflects real-world user interactions with map services while pushing state-of-the-art models to understand spatial relationships, map infographics, travel planning, POI search, and navigation. MapEval ensures geographic diversity, realistic query patterns, and evaluation across multiple modalities. By integrating long contexts, visual complexity, API interactions, and questions requiring commonsense reasoning or recognition of insufficient information (i.e., unanswerability), it offers a rigorous framework for advancing geo-spatial AI capabilities. In Fig 1, we depict an overview of MapEval.

With MapEval, we evaluated 30 prominent foundation models, where Claude-3.5-Sonnet, GPT-4o, and Gemini-1.5-Pro showed competitive performance overall. However, significant gaps emerged in MapEval-API, with Claude-3.5-Sonnet agents outperforming GPT-4o and Gemini-1.5-Pro by 16% and 21%, respectively, and even larger disparities compared to open-source models. Our detailed analyses revealed further insights into model strengths and weaknesses. Despite these advances, all models still fall short of human performance by over 20%, especially in handling complex map images and rigorous reasoning, underscoring MapEval's role in advancing geo-spatial understanding.

## 2. Related Work

Geo-spatial question answering presents significant challenges for foundation models (Mai et al., 2024). Early research in GeoQA (Mai et al., 2021) has focused on template-based methods (Zelle & Mooney, 1996; Chen et al., 2013; Chen, 2014; Punjani et al., 2018; Kefalidis et al., 2023), where predefined templates classify queries and retrieve information from structured databases like OpenStreetMap or DBpedia (Auer et al., 2007). While effective in certain scenarios, these methods are constrained by the static nature of the databases and the predefined templates, limiting their flexibility in handling complex or dynamic queries. There has been limited effort to assess (Roberts et al.) and improve (Balsebre et al., 2024) LLMs' capabilities in geospatial reasoning. Recent benchmarks such as Travel Planner (Xie et al.), ToolBench (Qin et al.), and API-Bank (Li et al., 2023) integrate map tools and APIs for location-based queries. While these benchmarks handle real-world tasks like itinerary planning or querying map data, the use of map APIs is limited to more straightforward use cases, such as calculating distances or identifying nearby points of interest. In addition, remote sensing research (Bastani et al., 2023; Yuan et al., 2024; Zhang et al., 2024; Lobry et al., 2020) has focused on extracting physical features from satellite imagery. While valuable for environmental monitoring and urban planning, this approach differs significantly from the task of reasoning over interactive digital map views, which

involve understanding spatial relationships, map symbols, and navigation elements in a dynamic, user-interactive context. For a detailed discussion of template-based approaches, benchmarks involving map APIs, and comparisons with remote sensing methodologies, see Appendix A.

## 3. The MapEval Dataset

### 3.1. Design Principles

**Reasoning.** Geo-spatial reasoning in map-based tasks presents distinct challenges for foundation models, including: (a) understanding complex problem descriptions in natural language, (b) collecting relevant world information using map tools or APIs, (c) performing compositional and spatio-temporal reasoning, (d) interpreting map visuals, and (e) synthesizing information from heterogeneous geo-spatial contexts (e.g., named entities, distances, and temporal values). These tasks test the limits of state-of-the-art models, which struggle to fully grasp geo-spatial relationships, navigation complexities, and POIs.

**Realistic.** MapEval reflects real-world map usage by capturing typical user interactions with map services, such as: (a) varied usage patterns like location-based searches and travel planning, and (b) informal, often fragmented queries, without relying on perfect grammar or structure.

**Diversity.** MapEval ensures geographic diversity and broad evaluation across models and tasks: (a) capturing locations across cities and countries globally, and (b) offering a wide variety of question types and contexts, which test foundation models' spatial, temporal, data retrieval, and visual reasoning abilities.

**Long Contexts, Multi-modality, API Interactions.** MapEval challenges models with: (a) long geo-spatial descriptions, including POIs and navigational data, (b) complex map-specific images with location markers, and (c) API interactions, testing models' abilities as language agents in real-world map-based tasks.

**Unanswerability, Commonsense.** MapEval includes questions where context is insufficient to provide an answer, testing models' ability to identify missing or incomplete information, rather than making incorrect guesses. It also assesses commonsense reasoning and handling uncertainty, essential for reliable decision-making in real-world applications.

**Multiple Choice Questions (MCQs).** We employ MCQs in MapEval, similar to MMLU (Hendrycks et al.), rather than open-ended queries. This approach circumvents the evaluation challenges associated with generated responses (Sai et al., 2022), allowing for a more straightforward and reliable accuracy-based assessment of map-based reasoning capabilities. As discussed in Appendix G.2, we also evaluate

an open-ended variant of MapEval to assess its flexibility beyond MCQ format.

**Cost-effective and Focused.** To maintain a cost-effective yet comprehensive evaluation framework, MapEval comprises 700 instances, deliberately designed to cover diverse geo-spatial reasoning tasks, question types, and global locations. This focused approach ensures a manageable testbed while providing sufficient diversity for robust and reliable assessment, avoiding redundancy without compromising depth or breadth.

### 3.2. Tasks

**Textual.** The objective of MapEval-Textual is to answer MCQs by decomposing complex queries and extracting relevant information from long textual contexts. These contexts describe map locations, POIs, routes, navigation details, and travel distances/times, often including user ratings or reviews. Unlike typical reading comprehension tasks, these texts combine structured data (e.g., coordinates, distances) with unstructured narratives and subjective content. The model must reason over this heterogeneous information to select the correct answer. This task evaluates the model's ability to analyze fine-grained map-related information presented in text.

**API.** In the MapEval-API task, an AI agent interacts with map-based APIs to retrieve data (e.g., nearby POIs, distance calculations). The task involves generating API queries based on user questions, interpreting the returned structured data, and integrating it into reasoning processes to answer MCQs. This task evaluates the model's ability to handle data retrieval, API interactions, and the synthesis of structured information in real-world, map-driven scenarios.

**Visual.** MapEval-Visual task requires the model to interpret and analyze map snapshots, specifically digital map views from services like Google Maps. These snapshots represent complex spatial relationships, routes, landmarks, OCR texts (e.g., rating), and symbolic elements (e.g., logos or traffic signs), which differ from typical image recognition tasks. The model must extract relevant information from the visuals, integrate it with spatial reasoning, and use it to answer MCQs. This task assesses the model's ability to tackle map-specific visual contents and perform spatial reasoning.

### 3.3. Dataset Construction

#### Data Annotation.

To create a high-quality benchmark dataset for MapEval, we utilized Google Maps, a widely adopted map service. The process of constructing the textual context presented significant challenges, particularly in ensuring accuracy and efficiency. For an example question like "What are the open-

| Type | Task | Question Example | Count |
|------|------|------------------|-------|
| Place Info | Textual/API | What is the direction of Victoria Falls from Harare? | 64 |
| | Visual | Is there any Hospital marked with a star symbol on the tourist map of Rome? | 121 |
| Nearby | Textual/API | Find restaurants nearby Louvre Museum above 4.0 rating. | 83 |
| | Visual | I stayed at SpringHill Suites by Marriott Portland Hillsboro. Can you recommend the nearest restaurant to my location? | 91 |
| Routing | Textual/API | I am driving to Brassica in Bexley Via E Whittier St. After reaching Lockbourne Rd, where should I go next? | 66 |
| | Visual | What is the fastest route from Times Square to Central Park by walking? | 80 |
| Unanswerable | Textual/API | Which road should I follow from Wola to Mokotów to avoid flooded roads in heavy rains? | 20 |
| | Visual | Which way should be efficient while visit from Abis bus station to KONO so that Victoria park is on the way | 20 |
| Trip | Textual/API | I have an afternoon free in New York and plan to visit The Metropolitan Museum of Art for 3 hours, followed by a 30-minute coffee break at a nearby cafe, and then spend 1 hour in Central Park. Plan a schedule to ensure I have enough time for everything. | 67 |
| Counting | Visual | How many hospitals are there in the left side of the river? | 88 |

*Table 1.* Examples of different question categories. MapEval-Textual and MapEval-Visual questions are accompanied by both textual and visual context (See appendix I for full qualitative example queries, contexts and evaluation model outputs during evaluations.)

ing hours of the British Museum?" requires precise data to provide valid options and a correct answer. Manually searching for the "British Museum" on Google Maps and looking for its opening hours can be both time-consuming and prone to errors, making this method inefficient. To address these challenges, we used MapQaTor (Dihan et al., 2024), a web interface built on Google Maps APIs, designed to streamline the collection of textual map data. MapQaTor automates data retrieval from map APIs, collecting key information like opening hours and location details to build the textual dataset (Details in Appendix B.1). For each user query, we first fetch the necessary context data using MapQaTor. Questions were then paired with their corresponding contexts, and multiple-choice options were carefully curated based on this information. The ground truth answers were derived from the same context.

For MapEval-API, the same questions were used as in MapEval-Textual, but without textual contexts, requiring the language agents to interact with tools directly. To address consistency issues with real-time data updates, we created a controlled evaluation environment. This involves caching place information and simulating API interactions. Details of the pseudo-Google Maps setup are provided in Appendix C.

For the visual context, we capture map snapshots from Google Maps, covering random locations across various cities and countries worldwide. Based on each snapshot, we formulate relevant questions with multiple-choice options, where the correct labels are derived directly from the map

information. To maintain traceability, we save the Google Maps URL for each snapshot. Additionally, to examine model capabilities at different zoom levels, we capture snapshots at varying zoom depths[1].

We create the following question types for MapEval: (a) Place Info: detects POIs and asks about specific details related to a place (e.g., location, rating, reviews); (b) Nearby: identifies nearby places or POIs; (c) Routing: navigates between locations, considering routes and landmarks; (d) Unanswerable: when the map information (e.g., from google map) or the textual and visual context is insufficient to answer the question. Note that, in each category we formulate a few questions that requires general knowledge or reasoning about locations and navigation (e.g., there are 52 commonsense QAs in MapEval-Visual).

Moreover, MapEval-Textual and MapEval-API exclusively feature Trip questions, which involve planning multi-stop journeys across various POIs. Due to the complexity and details of trip planning, these questions are difficult to represent in a single visual snapshot. Conversely, Counting tasks are unique to MapEval-Visual, where models count specific items or locations on a map—a challenge specifically tailored to visual contexts.

**Quality Control and Human Performance** To ensure quality, each QA pair is annotated by multiple members of our

---

[1]Zoom levels found in map URLs indicate depth (e.g., url has zoom level 16.71), with higher values (e.g., 16 and above) showing more detail, compared to level 1 (world map)- See Appendix F.1

| Statistics | Number |
|---|---|
| Total unique question instances | 700 |
| - Questions with api or textual-context | 300 (42.86%) |
| - Questions with visual-context | 400 (57.14%) |
| Total unique countries | 54 |
| Total unique cities | 180 |
| Maximum textual-context length | 1500 |
| Maximum question length | 107 |
| Average textual-context length | 435.63 |
| Average question length | 21.41 |
| Unique number of textual-context | 215 |
| Unique number of visual-context | 270 |
| Min, Max, Avg questions from a country | 6, 132, 12.96 |
| Min, Max, Avg questions from a city | 1, 44, 3.89 |
| Min, Max, Avg Choices | 2, 7, 4.004 |
| Max zoom of visual-context | 21.0 |
| Min zoom of visual-context | 8.0 |
| Average zoom of visual-context | 15.26 |

*Table 2.* Key statistics of MapEval. Lengths are in words. Visual-context means Map snapshots/images. Some questions are yes/no and some have additional complexity with 4+ choices.

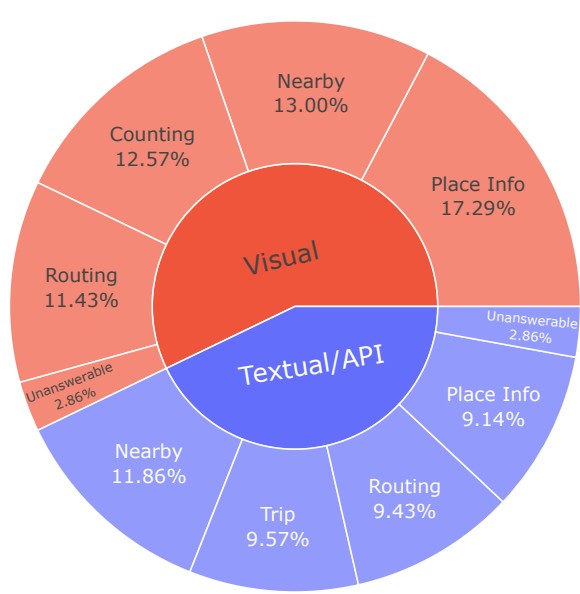

*Figure 2.* MapEval category statistics.

team, achieving an initial 76% mutual agreement. At least two team members then manually verify and resolve any disputes on the remaining pairs; if consensus cannot be reached (i.e., ambiguous), that pair is filtered out. To compute human scores, two team members who did not participate in the annotation process attempt to answer the questions, and their highest-scoring attempts are reported as the human performance benchmark. For MapEval-API, as the questions are identical to MapEval-Textual, we report the same human performance for both.

### 3.4. Dataset Statistics and Analysis

The main statistics of MapEval are presented in Table 2 and Figure 2. Examples of each question type and their numbers are presented in Table 15. We visualize the global distribution of locations in our dataset using coordinates (Fig. 7 in appendix). Table 12 (Appendix) lists all countries and their frequencies in MapEval. We use OpenStreetMap's Nominatim API for reverse geocoding to determine countries from coordinates. Textual context includes the coordinates of places in it. In case of visual context, we can find the coordinates from the associated Map URL with each snapshot. For example, coordinate of an example url, is 35.7048455,139.763263. We visualize the distribution of question and textual context lengths in the Appendix (Figures 5 and 6). Overall, beyond their diversity in types, questions and contexts also vary significantly in length, reflect-

ing varying levels of complexity and detail. Furthermore, in Appendix F.1, we illustrate the zoom level distribution in MapEval-Visual, adding another dimension to the dataset's diversity and evaluation challenges.

## 4. Experiments

### 4.1. Experimental Protocol and Setup

We evaluate all tasks using the accuracy metric, defined as the percentage of correct choices selected by the model. We prompt models with the respective context, question, tool usage documentations (only for MapEval-API), answer format guidelines, and choices. We assess LLMs for MapEval-Textual, VLMs for MapEval-Visual, and ReACT agents (Yao et al., 2023) (known for effective tool interaction (Zhuang et al., 2023)) built on various LLMs for MapEval-API, aligning each task with appropriate model types. Appendix I presents example prompts for all tasks. Our LLMs and VLMs spans both open and closed-source models. Closed-source models include Claude-3.5-Sonnet, GPT-4o, GPT-4-Turbo, GPT-3.5-Turbo, Gemini-1.5 (Pro, Flash), with all except GPT-3.5-Turbo being multi-modal foundation models used in all tasks, while GPT-3.5-Turbo, which is text-only, is utilized solely in the MapEval-Textual and MapEval-API tasks. Open-source LLMs include instruct versions of Gemma-2.0 (9B, 27B), Llama-3.2 (3B, 90B), Llama-3.1 (8B, 70B) , Mistral-Nemo-7B, Mixtral-8x7B,

| Model | Overall (#300) | Place Info (#64) | Nearby (#83) | Routing (#66) | Trip (#67) | Unanswerable (#20) |
|---|---|---|---|---|---|---|
| Close-Source (Proprietary) LLMs | | | | | | |
| Claude-3.5-Sonnet (Anthropic, 2024) | **66.33** | **73.44** | 73.49 | **75.76** | **49.25** | 40.00 |
| Gemini-1.5-Pro (Team et al., 2024a) | **66.33** | 65.63 | **74.70** | 69.70 | 47.76 | **85.00** |
| GPT-4o (OpenAI, 2024) | 63.33 | 64.06 | **74.70** | 69.70 | **49.25** | 40.00 |
| GPT-4-Turbo (OpenAI, 2023) | 62.33 | 67.19 | 71.08 | 71.21 | 47.76 | 30.00 |
| Gemini-1.5-Flash (Team et al., 2024a) | 58.67 | 62.50 | 67.47 | 66.67 | 38.81 | 50.00 |
| GPT-4o-mini (OpenAI, 2024) | 51.00 | 46.88 | 63.86 | 57.58 | 40.30 | 25.00 |
| GPT-3.5-Turbo (OpenAI, 2022) | 37.67 | 26.56 | 53.01 | 48.48 | 28.36 | 5.00 |
| Open-Source LLMs | | | | | | |
| Llama-3.1-70B (AI@Meta, 2024) | **61.00** | **70.31** | 67.47 | **69.70** | 40.30 | **45.00** |
| Llama-3.2-90B (AI@Meta, 2024) | 58.33 | 68.75 | 66.27 | 66.67 | 38.81 | 30.00 |
| Qwen2.5-72B (Team, 2024) | 57.00 | 62.50 | **71.08** | 63.64 | **41.79** | 10.00 |
| Qwen2.5-14B (Team, 2024) | 53.67 | 57.81 | **71.08** | 59.09 | 32.84 | 20.00 |
| Gemma-2.0-27B (Team et al., 2024b) | 49.00 | 39.06 | **71.08** | 59.09 | 31.34 | 15.00 |
| Gemma-2.0-9B (Team et al., 2024b) | 47.33 | 50.00 | 50.60 | 59.09 | 34.33 | 30.00 |
| Llama-3.1-8B (AI@Meta, 2024) | 44.00 | 53.13 | 57.83 | 45.45 | 23.88 | 20.00 |
| Qwen2.5-7B (Team, 2024) | 43.33 | 48.44 | 49.40 | 42.42 | 38.81 | 20.00 |
| Mistral-Nemo (AI, 2024) | 43.33 | 46.88 | 50.60 | 50.00 | 32.84 | 15.00 |
| Mixtral-8x7B (Jiang et al., 2024) | 43.00 | 53.13 | 54.22 | 45.45 | 26.87 | 10.00 |
| Phi-3.5-mini (Abdin et al., 2024) | 37.00 | 40.63 | 48.19 | 46.97 | 20.90 | 0.00 |
| Llama-3.2-3B (AI@Meta, 2024) | 33.00 | 31.25 | 49.40 | 31.82 | 25.37 | 0.00 |
| Human Performance | | | | | | |
| Human | 86.67 | 92.19 | 90.36 | 81.81 | 88.06 | 65.00 |

*Table 3.* MapEval-Textual performances. Figure 16 visualizes the categorical accuracy.

Qwen2.5 (7B, 14B, 72B) , Phi-3.5-mini. For MapEval-Visual, we considered the open-source VLMs: Qwen2.5-VL-72B-Instruct, Qwen2-VL-7B-Instruct, Llama-3.2-90B-Vision, MiniCPM-Llama3-V-2_5, Llama-3-VILA1.5-8B, glm-4v-9b, InternLm-xcomposer2, paligemma-3b-mix-224, DocOwl1.5, llava-v1.6-mistral-7b-hf, and llava-1.5-7b-hf. In MapEval-API task, we concentrate our exploration on high-capacity open-source LLMs, specifically Llama-3.2-90B, Llama-3.1-70B, Mixtral-8x7B, and Gemma-2.0-9B. We limit our evaluation of open-source models in AI agents due to the task's complexity and resource demands, the lower performance of smaller models, and the excessive number of calls for both LLMs and map APIs.

## 4.2. Results and Analysis

### 4.2.1. MAPEVAL-TEXTUAL

We present a summary of MapEval-Textual results in Table 3, highlighting insights into the current state of geo-spatial reasoning in language models. Closed-source models generally outperform open-source ones, with Claude-3.5-Sonnet achieving 66.33% accuracy, while the best open-source model, Llama-3.1-70B, reaches 61.00%. However, the substantial gap between top-performing models and human accuracy (86.67%) underscores the challenges in geo-

spatial reasoning tasks. Models perform well in "Place Info", "Nearby", and "Routing" tasks (best performance ~75%) due to the comprehensive context extracted by MapEval-Textual, such as textual descriptions, opening hours, and distances. In contrast, performance in "Trip" planning scenarios remains poor (best ~49%), reflecting difficulties in handling multi-step reasoning and aggregating spatio-temporal constraints. Performance on "Unanswerable" queries is mixed, with Gemini-1.5-Pro achieving 85% accuracy while most models fall between 0–45%. This disparity highlights the importance of recognizing insufficient information for real-world applications. Consistent underperformance in "Trip" tasks and varied accuracy in unanswerable queries point to fundamental limitations in geo-spatial reasoning across architectures The results also show that larger models generally outperform smaller ones. However, the gap between open and closed-source models indicates room for advancement in open-source systems. Additionally, Fig. 17 reveals challenges faced by open-source models in handling longer contexts.

### 4.2.2. MAPEVAL-API

We present the MapEval-API results in Table 4, highlighting key insights into the geo-spatial reasoning abilities of

| Model | Overall (#300) | Place Info (#64) | Nearby (#83) | Routing (#66) | Trip (#67) | Unanswerable (#20) |
|---|---|---|---|---|---|---|
| Close-Source (Proprietary) LLMs | | | | | | |
| Claude-3.5-Sonnet (Anthropic, 2024) | **64.00** | **68.75** | **55.42** | **65.15** | **71.64** | 55.00 |
| GPT-4-Turbo (OpenAI, 2023) | 53.67 | 62.50 | 50.60 | 60.61 | 50.75 | 25.00 |
| GPT-4o (OpenAI, 2024) | 48.67 | 59.38 | 40.96 | 50.00 | 56.72 | 15.00 |
| Gemini-1.5-Pro (Team et al., 2024a) | 43.33 | 65.63 | 30.12 | 40.91 | 34.33 | **65.00** |
| Gemini-1.5-Flash (Team et al., 2024a) | 41.67 | 51.56 | 38.55 | 46.97 | 34.33 | 30.00 |
| GPT-3.5-Turbo (OpenAI, 2022) | 27.33 | 39.06 | 22.89 | 33.33 | 19.40 | 15.00 |
| GPT-4o-mini (OpenAI, 2024) | 23.00 | 28.13 | 14.46 | 13.64 | 43.28 | 5.00 |
| Open-Source LLMs | | | | | | |
| Llama-3.2-90B (AI@Meta, 2024) | **39.67** | **54.69** | **37.35** | 39.39 | 35.82 | 15.00 |
| Llama-3.1-70B (AI@Meta, 2024) | 37.67 | 53.13 | 32.53 | **42.42** | 31.34 | 15.00 |
| Mixtral-8x7B (Jiang et al., 2024) | 27.67 | 32.81 | 18.07 | 27.27 | **38.81** | 15.00 |
| Gemma-2.0-9B (Team et al., 2024b) | 27.00 | 35.94 | 14.46 | 28.79 | 26.87 | **45.00** |

*Table 4.* MapEval-API evaluation performance (See Figure 18 to visualize categorical accuracy)

language models with map APIs. MapEval-API generally underperforms compared to MapEval-Textual , with significant drops in Nearby tasks (from 74.70% to 55.42%) and Routing tasks (from 75.76% to 65.15%). Figure 3 visualizes these differences. While Claude-3.5-Sonnet performed consistently, other models showed declines due to lack of context and tool complexity, emphasizing the need for advanced agents surpassing ReAct's capabilities. In the Trip category, MapEval-API improved by approximately 22% compared to MapEval-Textual, demonstrating its effectiveness in step-by-step reasoning for complex problems. Claude-3.5-Sonnet led with 64.00% overall accuracy, excelling as a tool agent and in generic graph reasoning. However, a significant gap exists between closed-source and open-source models, with the best open-source model, Llama-3.2 90B, achieving 39.67%. Performance on "Unanswerable" queries varied widely (5% to 65%), highlighting the need for better handling of insufficient information.

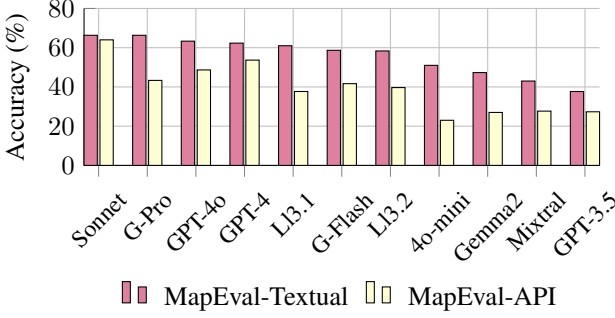

*Figure 3.* Comparison between MapEval-Textual and MapEval-API .

### 4.2.3. MAPEVAL-VISUAL

We evaluate models on the MapEval-Visual task in Table 5. Closed-source models outperform open-source ones, with Claude-3.5-Sonnet leading at 61.65%, followed by GPT-4o (58.90%) and Gemini-1.5-Pro (56.14%). Among open-source models, Qwen2.5-VL-72B achieves the highest accuracy (60.35%). While models excel in Place Info tasks (82.64%), they struggle with complex tasks like Counting, Nearby, and Routing, highlighting the need for improvement. Models trained on generic images underperform on map-specific tasks, likely due to limited exposure to detailed map data. Fig 21 (Appendix) shows accuracy dropping significantly at higher zoom levels (e.g., streets, symbols, demarcations) beyond level 14, where map complexity increases. Our benchmark reveals a substantial gap between AI and human performance, particularly in nuanced tasks. For example, humans achieve 85.18% on Routing tasks versus 50% for the best model, and 78.41% on Counting tasks versus 47.73%. While closed-source models like Claude-3.5-Sonnet and Gemini-1.5-Pro excel in identifying unanswerable questions (90% and 80%, respectively), open-source models struggle significantly.

### 4.3. Qualitative Error Analysis

LLMs struggle with spatial, temporal, and commonsense reasoning in location-based queries. In spatial reasoning, they often fail with straight-line distances (Listing 1), cardinal directions (e.g., East, West, North, South; Listing 2), and step-by-step route planning, including tasks that involve mathematical computations or counting (e.g., nearby restaurant counts; Listing 3). Temporal reasoning challenges include inefficient trip planning and errors in travel times or visit durations (Listing 4). Commonsense reasoning failures arise when models hallucinate or miss simple contextual

| Model | Overall (#400) | Place Info (#121) | Nearby (#91) | Routing (#80) | Counting (#88) | Unanswerable (#20) |
|---|---|---|---|---|---|---|
| Close-Source (Proprietary) VLMs | | | | | | |
| Claude-3-5-Sonnet (Anthropic, 2024) | **61.65** | **82.64** | 55.56 | 45.00 | **47.73** | **90.00** |
| GPT-4o (OpenAI, 2024) | 58.90 | 76.86 | **57.78** | **50.00** | **47.73** | 40.00 |
| Gemini-1.5-Pro (Team et al., 2024a) | 56.14 | 76.86 | 56.67 | 43.75 | 32.95 | 80.00 |
| GPT-4-Turbo (OpenAI, 2023) | 55.89 | 75.21 | 56.67 | 42.50 | 44.32 | 40.00 |
| Gemini-1.5-Flash (Team et al., 2024a) | 51.94 | 70.25 | 56.47 | 38.36 | 32.95 | 55.00 |
| GPT-4o-mini (OpenAI, 2024) | 50.13 | 77.69 | 47.78 | 41.25 | 28.41 | 25.00 |
| Open-Source VLMs | | | | | | |
| Qwen2.5-VL-72B (Bai et al., 2025) | **60.35** | 76.86 | **54.44** | 43.04 | **52.33** | **90.00** |
| Qwen2-VL-7B (Wang et al., 2024) | 51.63 | 71.07 | 48.89 | 40.00 | 40.91 | 40.00 |
| Llama3.2-90B-Vision (AI@Meta, 2024) | 50.38 | 73.55 | 46.67 | 41.25 | 36.36 | 25.00 |
| Glm-4v-9b (GLM et al., 2024) | 48.12 | 73.55 | 42.22 | 41.25 | 34.09 | 10.00 |
| InternLm-Xcomposer2 (Dong et al., 2024) | 43.11 | 50.41 | 48.89 | **43.75** | 34.09 | 10.00 |
| MiniCPM-Llama3-V-2_5 (Yao et al., 2024) | 40.60 | 60.33 | 32.22 | 32.50 | 31.82 | 30.00 |
| Llama-3-VILA1.5-8B (Lin et al., 2023) | 32.99 | 46.90 | 32.22 | 28.75 | 26.14 | 5.00 |
| Llava-v1.6-Mistral-7B-hf (Liu et al., 2024b) | 31.33 | 42.15 | 28.89 | 32.50 | 21.59 | 15.00 |
| DocOwl1.5 (Hu et al., 2024) | 31.08 | 43.80 | 23.33 | 32.50 | 27.27 | 0.00 |
| Paligemma-3B-mix-224 (Beyer et al., 2024) | 30.58 | 37.19 | 25.56 | 38.75 | 23.86 | 10.00 |
| Llava-1.5-7B-hf (Liu et al., 2024a) | 20.05 | 22.31 | 18.89 | 13.75 | 28.41 | 0.00 |
| Human Performance | | | | | | |
| Human | 82.23 | 81.67 | 82.42 | 85.18 | 78.41 | 65.00 |

*Table 5.* MapEval-Visual evaluation performance. (Fig 20 visualizes categorical accuracy).

deductions (Listing 5). LLM-based agents also struggle with map tools and APIs, especially in Nearby and Routing queries. Incorrect parameter usage results in failures, such as missing key parameters, using incompatible values, or entering infinite request loops when no valid results are found. Visual tasks reveal further issues, with VLMs struggling to maintain spatial awareness, misidentifying closely located POIs, or incorrectly counting POIs in map images (e.g., malls/stores). These limitations highlight the need for better spatial awareness, temporal reasoning, and robust tool usage (details Appendix E).

## 5. Addressing Failures and Enhancing Geospatial Reasoning in LLMs

**Failures in MapEval-Textual:** To further analyze LLMs' performance in MapEval-Textual, we examined a subset of questions requiring (i) straight-line distance calculations (47 questions), (ii) determining cardinal directions (24 questions), and (iii) counting-related queries (23 questions). The results revealed significant variability in model performance across these tasks (Fig. 11, Fig. 12, Fig. 13): (i) Claude-3.5-Sonnet achieved the highest accuracy (91%) in identifying cardinal directions, while Gemma-2.0-27B scored the lowest (16.67%). (ii) Straight-line distance calculations posed challenges for all models, with the best accuracy at only 51.06%. (iii) Counting tasks also proved difficult, with

| Model | Straight-Line Distance (#47) | | Cardinal Direction (#24) | |
|---|---|---|---|---|
| | LLM | LLM+ Calculator | LLM | LLM+ Calculator |
| Close-Source (Proprietary) LLMs | | | | |
| Claude-3.5-Sonnet | 51.06 | 85.11 | 91.67 | 95.83 |
| GPT-4o | 46.81 | 70.21 | 62.50 | 87.50 |
| GPT-4-Turbo | 40.43 | 76.59 | 58.33 | 91.67 |
| Gemini-1.5-Pro | 38.29 | 72.34 | 62.50 | 91.67 |
| Gemini-1.5-Flash | 46.81 | 63.83 | 58.33 | 87.50 |
| GPT-4o-mini | 34.04 | 78.72 | 29.17 | 91.67 |
| GPT-3.5-Turbo | 19.15 | 55.32 | 20.83 | 62.50 |
| Open-Source LLMs | | | | |
| Llama-3.2-90B | 42.55 | 68.90 | 66.67 | 87.50 |
| Llama-3.1-70B | 48.94 | 61.7 | 66.67 | 95.83 |
| Mixtral-8x7B | 38.29 | 59.57 | 33.33 | 79.17 |
| Gemma-2.0-9B | 29.79 | 68.09 | 37.50 | 75.00 |

*Table 6.* Performance Improvement of LLMs (MapEval-Textual) in Straight-Line Distance and Cardinal Directions Analysis (Fig. 14 and 15 visualizes the improvement).

even Claude-3.5-Sonnet underperforming compared to the open-source Gemma-2.0-27B (60.87% accuracy).

**Calculator Integration for Complex Spatial Computations:** These results suggest that LLMs struggle with spatial reasoning tasks that involve precise numerical computations. To mitigate these limitations, we extended model capabilities by integrating external tools (e.g., calculators) for tasks like calculating distances and determining cardi-

nal directions (details in Appendix G). This led to dramatic improvements (Table 6): (i) Claude-3.5-Sonnet's accuracy in calculating straight-line distances increased from 51.06% to 85.11%. (ii) GPT-4o-mini, which initially struggled with cardinal directions, improved from 29.17% to 91.67%. (iii) Even open-source models like Gemma-2.0-9B saw accuracy boosts in straight-line distance tasks, rising from 29.79% to 68.90%. These results highlight the challenges LLMs face in spatial reasoning without external support. Integrating tools allows models to offload computationally intensive tasks, yielding more precise and reliable results. However, spatial reasoning is just one aspect of location-based tasks where models underperform. Temporal reasoning tasks, such as accounting for travel times or determining optimal visiting hours, could similarly benefit from tool integration. Expanding tool support could enhance performance across multiple domains but would also increase architectural complexity, requiring effective management of diverse tools.

**Failures in MapEval-API:** In ReAct-based systems, a significant challenge arises from the heavy responsibility placed on a single agent to extract relevant parameters from a question, call APIs with those parameters, and then provide the final answer based on API responses. This complex process often leads to issues such as parameter extraction errors, incorrect API calls, or dead loops (e.g., GPT-3.5-Turbo encountering 16 infinite iterations; see Fig. 19). These problems become more pronounced when the agent struggles with reasoning through the task, reducing task completion rates. Moreover, processing large volumes of API data, even in plain text form (i.e., long contexts in MapEval-Textual task), poses a significant challenge for LLMs, as discussed in Section 4.2.1 and reflected in the low performances in Table 3.

**Adaptive Routing of Tools and Models:** To address these limitations, the Chameleon Framework (Lu et al., 2024) offer a robust solution by adaptively breaking tasks into multiple tool-usage modules (e.g., multi-agent systems). The

| Category | ReAct | Chameleon |
|---|---|---|
| Place Info | 39.06 | 54.69 |
| Nearby | 22.89 | 54.21 |
| Routing | 33.33 | 51.51 |
| Trip | 19.40 | 43.28 |
| Unanswerable | 15.00 | 25.00 |
| Overall | 27.33 | 49.33 |

*Table 7.* Performance improvement of GPT-3.5-Turbo in MapEval-API

integration of Chameleon into MapEval-API has already demonstrated a notable improvement in GPT-3.5-Turbo's performance (Table 7). Moreover, Chameleon's ability to decompose tasks and handle errors more efficiently results in fewer parameter extraction errors and prevents dead loops,

significantly boosting accuracy. Another promising alternative is developing an ensemble system that combines a query classifier with type-specific LLM deployment. This system would first classify incoming queries and then route them to the best-performing LLM for that particular query type, potentially achieving superior performance.

## 6. Conclusion

In this paper, we introduce MapEval, a benchmark dataset designed to assess foundation models in geo-spatial reasoning through *textual*, *API-based*, and *visual* evaluation modes. MapEval incorporates diverse real-world scenarios to evaluate model capabilities on geo-spatial reasoning tasks. Our findings reveal that while leading models like Claude-3.5-Sonnet, GPT-4o, and Gemini-1.5-Pro excel in certain areas, they still underperform compared to human accuracy, especially when using open-source foundation models. This highlights critical areas for improvement, particularly in managing complex map-based queries requiring multi-step spatio-temporal reasoning, efficient tool utilization, and domain-specific knowledge. Future work could focus on developing specialized geospatial models, integrating LLMs with external tools like map APIs, and enhancing VLMs' visual understanding of map images. We anticipate that MapEval will catalyze ongoing research in geospatial reasoning and broader QA domains.

## Impact Statement

We ensure the reproducibility of our results by providing the complete dataset and evaluation codes used in our experiments[2][3][4]. These resources include the inference process for LLMs, detailing parameters such as temperature, top-k, and top-p. Any updates or bug fixes will be made available in the repository to maintain long-term usability.

Despite these efforts, our study has several limitations. The dataset focuses on five Google Maps APIs from the Places and Routes categories—Text Search, Place Details, Nearby Search, Directions, and Distance Matrix—excluding other categories like Maps and Environment. This limits the diversity of queries and geospatial insights evaluated. Additionally, updates to these APIs may render the dataset less relevant for real-time applications, making it more suitable for archival purposes.

The generalizability of our findings to other domains or tools remains untested, and variations in prompt formulations, which may influence results, were not explored. Future

---

[2] https://github.com/MapEval/MapEval-Textual/
[3] https://github.com/MapEval/MapEval-API/
[4] https://github.com/MapEval/MapEval-Visual/

research could address these gaps by incorporating a broader range of APIs, examining the impact of prompt design, and testing the methods across diverse domains.

By acknowledging these limitations and sharing our resources, we aim to support ongoing research and encourage further exploration of geospatial reasoning tasks using LLMs.

## Acknowledgment

We gratefully acknowledge the support of Qatar Computing Research Institute (QCRI) for providing access to APIs and computational resources, including GPU support, which were instrumental in conducting this research.

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

# A. Detailed Related Work

## A.1. MapEval-Textual

Template-based GeoQA models (Zelle & Mooney, 1996; Chen et al., 2013; Chen, 2014; Punjani et al., 2018; Kefalidis et al., 2023) have predominantly followed a two-step strategy for answering geographic questions: (1) classifying a natural language query into predefined templates and (2) using these templates to query structured geographic knowledge sources such as PostGIS, DBpedia (Auer et al., 2007), YAGO (Suchanek et al., 2007), Freebase (Bollacker et al., 2007), and OpenStreetMap. While these approaches are effective for structured queries, they are limited by the predefined question templates and their reliance on static databases. They typically convert natural language questions into structured query language scripts. For instance, GeoQuestions1089 (Kefalidis et al., 2023) contains 1089 questions with corresponding GeoSPARQL (Open Geospatial Consortium, 2011) queries over the YAGO2geo (Karalis et al., 2019) geospatial knowledge graph.

In contrast, our MapEval-Textual approach shifts the focus from database querying to assessing geospatial reasoning in Large Language Models (LLMs). Annotators collect factual map services data using MapQaTor, which is then provided as context to LLMs. This setup isolates and evaluates the model's ability to reason over geospatial relationships, addressing the challenge of free-form, complex map-related queries in a dynamic environment. This approach allows for a more holistic evaluation of LLMs, reflecting real-world usage where users interact with map tools using natural language queries. Thus, in MapEval, the responsibility lies with LLMs to answer the questions, whereas in previous works, the models were tasked with generating queries (e.g., Geoquery, GeoSPARQL), which are used to query external knowledge bases.

GPT4GEO (Roberts et al.) explored GPT-4's factual geographic knowledge by characterizing what it "knows" about the world without plugins or Internet access. Their evaluation focused on analyzing a single model using templated queries about generic location and direction-oriented facts, such as routing, navigation, and planning for well-known cities and places. However, this approach is inherently constrained by the training data of GPT-4, making it incapable of answering questions about less-known places. While the findings suggest that GPT-4 shows promising geo-spatial knowledge, this approach neither establishes a benchmark for geo-spatial reasoning nor incorporates real-life user queries or map services (e.g., Google Maps) as a geospatial information base.

Our approach employs fundamentally different evaluation and design principles. We establish a benchmarking of deeper geo-spatio-temporal reasoning capabilities across multiple foundation models using real user queries rather than templates. Uniquely, our evaluation encompasses multimodal understanding, tool interactions, and answerability determination. Additionally, we provide foundation models with fine-grained map services data through both context and API access, enabling a more comprehensive benchmarking of their geospatial question-answering abilities.

## A.2. MapEval-API

The MapEval-API task adopts a practical approach by leveraging map APIs to answer location-based questions directly, providing a more real-world scenario for evaluating the capabilities of Large Language Models (LLMs) in map-based reasoning. Recent advancements in LLMs have led to growing interest in planning tasks (Xie et al.; Balsebre et al., 2024; Zheng et al., 2024; Fang et al., 2024) that involve map data. For instance, the Travel Planner (Xie et al.) benchmark assessed multi-day itinerary planning using Google Maps API to determine distances, travel times, and details of nearby attractions. This task demonstrated the utility of map data in real-world planning scenarios, highlighting the potential for LLMs to integrate real-time geospatial information into decision-making.

Additionally, tool-calling benchmarks such as ToolBench (Qin et al.) and API-Bank (Li et al., 2023) have included location-based queries as a subtask, testing the ability of LLMs to interact with APIs in structured ways. These benchmarks typically focus on simpler query types, such as retrieving distances or nearby points of interest (POIs), but they do not fully address the complexity and diversity of real-world map-based questions.

In contrast, MapEval-API pushes the boundaries by evaluating LLMs on a wide variety of complex geospatial tasks that require not only querying map APIs but also integrating multiple pieces of information, such as travel itineraries, nearby services, and spatio-temporal reasoning. This more comprehensive evaluation of API-based reasoning challenges the models to process complex, multi-faceted questions, highlighting their ability to handle nuanced map interactions and effectively

synthesize data retrieved from APIs.

### A.3. MapEval-Visual

Prior works in geospatial analysis and map-based question answering have predominantly focused on remote sensing images (Bastani et al., 2023; Yuan et al., 2024; Zhang et al., 2024), which involve satellite or aerial imagery. These images often contain complex data about the Earth's surface, including land cover, vegetation, urban infrastructure, and other environmental features. Models designed for interpreting remote sensing images (Lobry et al., 2020) typically rely on convolutional neural networks (CNNs) and other computer vision techniques for object detection, segmentation, and classification tasks. These methods often focus on identifying physical entities like roads, buildings, and natural features from high-resolution imagery.

In contrast, our MapEval-Visual approach focuses on digital map view snapshots, which are 2D representations of map services (such as Google Maps). Unlike remote sensing images, which represent physical realities captured from a top-down perspective, these digital maps show geospatial information in a structured, interactive format. The focus of MapEval-Visual is to evaluate a model's ability to interpret and reason about these structured map views, which include not just physical features, but also symbolic and navigational information such as traffic signs, routes, landmarks, and visual cues from the map interface itself.

While remote sensing image analysis typically involves extracting physical data from raw image pixels, MapEval-Visual requires models to engage with spatial reasoning and map-based symbols, demanding a different set of computational skills. In this task, the model must not only understand the spatial relationships between map features but also reason about the context provided by digital map interfaces, which include additional elements such as zoom levels, icons, and navigation markers. This distinction sets MapEval-Visual apart from traditional remote sensing tasks and presents new challenges in the field of geospatial reasoning and map-based visual question answering.

## B. Data Collection Details

### B.1. MapQaTor: Annotator Interface

For the creation of the textual contexts and design MCQs based on that, we employed a custom-built web interface named **MapQaTor**. As illustrated in Figure 4, this interface was central to the dataset development process, offering an intuitive, user-friendly environment that simplifies complex tasks, such as API interaction and context generation.

The annotator interface is designed to reduce technical complexity for users, allowing them to concentrate on the core aspects of dataset annotation, such as selecting relevant locations, providing information on distances, durations, and directions between places, as well as identifying nearby points of interest. Its streamlined workflow facilitates efficient dataset creation by automating repetitive tasks, which not only minimizes errors but also significantly accelerates the annotation process.

MapQaTor uses five key Google Maps APIs: Text Search, Place Details, Distance Matrix, Directions, and Nearby Search, based on their relevance to common map-based tasks and their ability to provide comprehensive location data.

MapQaTor caches all API call responses, creating a static database for evaluation purposes. This ensures consistent responses when evaluating MapEval-API. Specifically, when an API call is made, the cached response is returned instead of a real-time query, maintaining a controlled and static evaluation environment.

Once the dataset is generated, it can be easily exported in JSON format, making it readily usable for further analysis and evaluation in downstream tasks, such as model training and benchmarking.

### B.2. Filtering via LLMs:

To ensure the challenge and quality of our dataset, we evaluated a range of LLMs. We filtered out samples where the majority of the LLMs could easily provide the correct answer, considering these samples "too easy" and removing them from the dataset. Additionally, we identified samples where most LLMs failed to answer the questions based on the given context. In such cases, we re-examined the questions, correcting any inconsistencies to improve clarity and relevance.

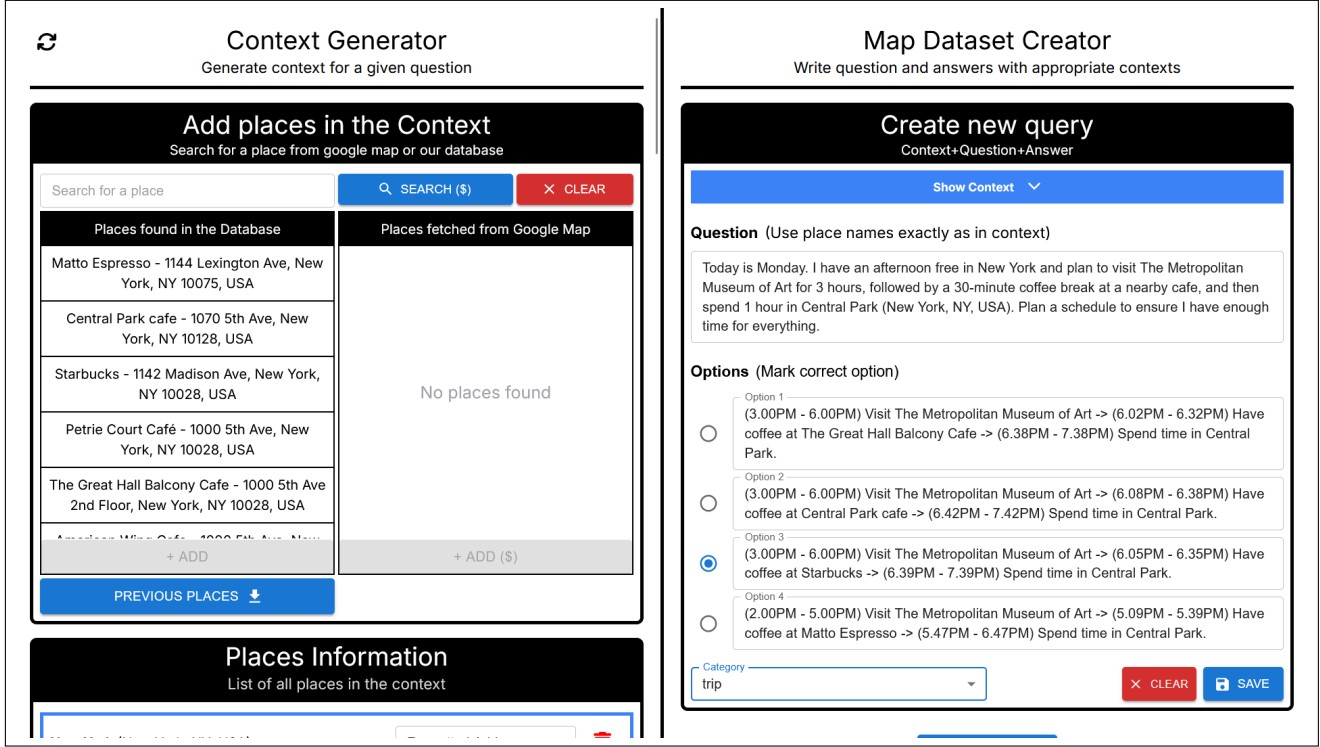

*Figure 4.* Screenshot of our Annotator Interface: MapQaTor

## C. Evaluation Details

### C.1. Pseudo-Google Maps Environment

To ensure consistency between annotation and evaluation, a pseudo-Google Maps environment was developed with the following features:

- Caching: Information for over 13,000 locations was cached using Google Maps place_ids during both annotation and evaluation stages, ensuring consistency across updates. Table 8 presents the number of data entries for each API tool in our database

- API Simulation: A proxy interface mimics actual API interactions, enabling controlled testing while maintaining dynamic map-like attributes (e.g., travel times and place lists).

- Key-Query Mapping: Discrepancies between user queries and database keys were handled by storing all data using standardized place_ids obtained via a real API call.

This method maintains a static evaluation environment to preserve answer validity while simulating real-world API interactions by controlling dynamic variables like travel times, place attributes, and nearby location lists, which often change in live settings.

### C.2. MapEval-Textual Evaluation

In this evaluation setting, we provide the LLM with a pre-fetched context containing detailed information about specific locations, such as opening hours, distances between points of interest, and nearby amenities. The context is designed to simulate a real-world scenario.

Listing 4 demonstrates an example of this evaluation process. The context includes details about The Metropolitan Museum of Art, including its location, opening hours, and nearby cafes. The query asks for a time-optimized schedule that includes a 3-hour visit to the museum, followed by a 30-minute coffee break at a nearby cafe, and 1 hour spent in Central Park.

The available options offer different schedules, and the models are tasked with selecting the most appropriate one based on the provided context. As illustrated, models like Claude-3.5-Sonnet, Gemini-1.5-Pro, and GPT-4o correctly identify Option 3 as the best fit, considering the opening hours of each location and the feasible travel times between them. In contrast, Gemma-2.0-9B selects an incorrect option, indicating a misunderstanding of the cafe's closing hours.

This pre-fetched context evaluation allows us to test the model's ability to reason over structured information and make contextually informed decisions. It highlights the importance of understanding spatial relationships, operating hours, and timing constraints, all of which are crucial in real-world trip planning tasks.

| Tool | Entries (#) |
|---|---|
| PlaceDetailsTool | 13,354 |
| TravelTimeTool | 1,142 |
| DirectionsTool | 317 |
| NearbySearchTool | 481 |

*Table 8.* Number of data entries in the database for each API tool

## C.3. MapEval-API Evaluation

In this evaluation approach, we leverage a Zero Shot React Agent, which utilizes a dynamic tool-based framework to enhance the model's ability to respond to user queries effectively. Listing 6 illustrates the structured system prompt guiding the agent in employing various available tools. This framework allows the agent to access a range of functionalities, including retrieving place IDs, obtaining detailed information about locations, and estimating travel times between points of interest.

The Zero Shot React Agent's dynamic capabilities enable it to interact with tools in a systematic manner, ensuring accurate and contextually relevant responses. For example, the agent can utilize the PlaceId tool to obtain the unique identifier for a specified location, which can then be employed in subsequent actions, such as fetching detailed information with PlaceDetails or finding nearby places using NearbyPlaces. This modular approach not only simplifies complex queries but also grounds responses in real-time data.

The prompt structure encourages the agent to think critically about each step, starting with the user's question and leading to a carefully considered action. It determines the appropriate tool to use, specifies the necessary input, and provides a well-structured JSON blob for the action. The observation of the tool's output informs the next steps, allowing for iterative refinement of the response.

By employing this Zero Shot React Agent framework, we can assess the model's proficiency in utilizing external tools to generate accurate, contextually aware responses, ultimately enhancing its effectiveness in real-world applications.

## C.4. MapEval-Visual Evaluation

In this scenario, we provide Large Language Models (LLMs) with a map snapshot that offers critical geospatial context necessary for answering the query. This snapshot includes a default map view with clearly labeled locations and roads, aiding in the understanding of spatial relationships.

The evaluation process is demonstrated in Listing 9. The input consists of two parts: an image as visual context and a corresponding query with multiple answer options. In the example provided, the visual context includes a map displaying several golf clubs and a complex roadway network. The options represent possible answers that the evaluated models can choose from. The listing also shows the responses of various models, along with their explanations.

Models like Gemini-1.5-Pro, GPT-4o-mini, and Claude-3.5-Sonnet successfully answered the query by correctly interpreting the geospatial information. On the other hand, Qwen-2VL-Chat selected an incorrect option, highlighting its difficulty in understanding spatial distances, which led to an erroneous answer.

This evaluation underscores the importance of providing visual context when testing LLMs' geospatial reasoning capabilities. By leveraging such visual aids, we can better assess how well these models understand and process spatial relationships in real-world scenarios

## D. Foundation Models' Details

Tables 10 and 11 provide comprehensive details of the open-source models utilized for dataset evaluation.

| Tool Name | Parameters | Description |
|---|---|---|
| PlaceSearch | placeName, placeAddress | Given a place name with address our tool calls Text Search API to get a list of places. Then choose the top place among them and returns its place id. |
| PlaceDetails | placeId | Given a place id our tool first searches in our database if not found, then uses Place Details API to fetch the details of the place. |
| TravelTime | originId, destinationId, travelMode | Given the place id of origin and destination, and travel mode our tool first searches in our database the duration (+distance) to go from origin to destination by preferred travel mode. If not found then queries Distance Matrix API. |
| Directions | originId, destinationId, travelMode | Given the place id of origin and destination, and travel mode our tool first searches in our database the available routes to go from origin to destination by preferred travel mode. If not found then queries Directions API. |
| NearbySearch | location, type, rankby, radius | This tool requires the place id of the place around which to retrieve place information. Additionally, the type of places, the order in which results are listed and distance within which to return place results. It then searches the database for stored Nearby Places. If absent, it queries Nearby Search API. |

*Table 9.* Summary of the tools used in evaluation.

## E. Fine-grained Qualitative Error Analysis

### E.1. MapEval-Textual

**Commonsense Reasoning**: (i) Consider a scenario where the context states, "{Place_A} serves dinner, lunch, vegetarian food." When asked, "Does {Place_A} serve breakfast?" many LLMs respond, "There is not enough information in the context to answer," instead of simply saying "No." A human would deduce that since breakfast is not listed, {Place_A} does not serve it. (ii) Another challenge arises in planning questions. Even when opening hours are included in the context, LLMs may plan schedules that overlook constraints, such as visiting during closed hours while satisfying other conditions. For instance, although {Place_A} is open from 9:00AM to 3:00PM, the model might schedule a visit at 5:00PM, possibly due to inadequate training for this scenario.

**Spatial Reasoning:** (i) LLMs particularly struggle with queries requiring the calculation of spatial relationships, such as cardinal directions, straight-line distances, nearest points of interest (POIs), or step-by-step route planning. For example, in Place Info, Nearby, and Routing questions, examining 50 random questions that required such computations we observed a 10% decreased accuracy than others. This decline highlights the limitations of even dominant models like Gemini, which struggle with straight-line distance and direction calculations from geo-spatial data. (ii) LLMs also encounter difficulty with our domain specific questions that involve maths even in counting, especially when the count is large. For instance, in a query like "How many nearby restaurants have at least a 4.5 rating?", LLMs often fail to provide an accurate count.

**Temporal Reasoning:** LLMs struggle with temporal reasoning, which affects their performance on tasks like trip planning that require time manipulation. For example, when asked, "I want to visit A, B, and C. What is the most efficient order to visit?" the model must calculate travel times and determine the optimal route but often fails. Similarly, in a query like, "I want to visit A for 1 hour. What is the latest time I can leave home?" the model needs to subtract the visit duration and travel time from A's closing time, yet frequently makes errors in these simple time calculations.

### E.2. MapEval-API

**Incorrect Tool Usage by Agents:** LLM-based agents often exhibit varying degrees of errors when utilizing map tools/APIs, particularly impacting Nearby queries. This task requires a complex set of arguments, and misinterpretation or improper use of these parameters frequently leads to failures in retrieving accurate results.

| Model | Parameters | Context Window |
|---|---|---|
| Phi-3.5-mini-instruct | 3.8B | 128K |
| Mistral-Nemo-Instruct-2407 | 7B | 128k |
| Mixtral-8x7B-Instruct-v0.1 | 7B | 32K |
| Qwen2.5-7B-Instruct | 7B | 128K |
| Qwen2.5-14B-Instruct | 14B | 128K |
| Qwen2.5-72B-Instruct | 72B | 128K |
| Llama-3.1-8B-Instruct | 8B | 128k |
| Llama-3.1-70B-Instruct | 70B | 128k |
| Llama-3.2-3B-Instruct | 3B | 128k |
| Llama-3.2-90B-text-preview | 90B | 128k |
| gemma-2-27b-it | 27B | 8.2k |
| gemma-2-9b-it | 9B | 8.2k |

*Table 10.* LLM model scales.

| Model | Parameters | Context Window |
|---|---|---|
| Qwen2.5-VL-72B-Instruct | 72B | 32k |
| Llama3.2-90B-Vision | 90B | 128k |
| MiniCPM-Llama3-V-2_5 | 7B | 8.2k |
| Qwen2-VL-7B-Instruct | 8B | 32K |
| Llama-3-VILA1.5-8B | 8B | 8.2k |
| glm-4v-9b | 4.9B | 100k |
| InternLm-xcomposer2 | 7B | 96K |
| paligemma-3b-mix-224 | 3B | - |
| DocOwl1.5 | 8B | - |
| llava-v1.6-mistral-7b-hf | 7B | - |
| llava-1.5-7b-hf | 7B | - |

*Table 11.* VLM model scales.

**Agents Stuck in Infinite Loops:** Invalid actions and repetitive loops contribute significantly to errors, especially in Routing queries. When there are no valid routes between an origin and destination, agents often fail to reconsider their approach or stop the process. Instead, they repeatedly attempt the same query with the same parameters, resulting in a deadlock and preventing progress.

### E.3. MapEval-Visual

**Spatial Reasoning:** In the Nearby category, models often exhibit confusion when multiple POIs are visually close together, leading to incorrect location selections. This indicates a struggle with fine-grained spatial analysis, affecting their ability to provide reliable responses and emphasizing the need for improved spatial awareness mechanisms.

**Temporal Reasoning:** In Routing queries, determining the fastest route requires detailed analysis of the source and destination, as well as transportation paths. VLMs often struggle with these calculations, resulting in a noticeable decline in performance and underscoring the difficulties in processing geographical information effectively.

**Detecting and Counting:** Models often struggle to accurately identify and count POIs in map images. For instance, when asked, "How many shopping stores or malls are there?" many proprietary VLMs may count incorrectly, with Claude-Sonnet listing an ATM as a store, leading to overcounting. Conversely, they sometimes undercount, (e.g., detecting only 6 to 8 out of an actual 12 malls).

## F. Dataset Statistics and Analysis

| Country | Count | Country | Count | Country | Count |
|---|---|---|---|---|---|
| Bangladesh | 132 | United States | 57 | United Arab Emirates | 40 |
| India | 33 | Canada | 31 | United Kingdom | 27 |
| Japan | 24 | Australia | 19 | Pakistan | 16 |
| Qatar | 15 | Saudi Arabia | 12 | China | 12 |
| Germany | 10 | Argentina | 10 | Luxembourg | 9 |
| Italy | 8 | Spain | 8 | Brazil | 8 |
| South Africa | 7 | Poland | 7 | New Zealand | 7 |
| France | 7 | Denmark | 7 | Bhutan | 7 |
| Hungary | 7 | Czechia | 7 | Chile | 7 |
| Sierra Leone | 7 | Malaysia | 7 | Sweden | 7 |
| Norway | 7 | Peru | 6 | Colombia | 6 |
| Zimbabwe | 6 | Ireland | 6 | Mexico | 6 |
| Egypt | 6 | Greece | 6 | Austria | 6 |
| Indonesia | 6 | Nepal | 6 | Netherlands | 6 |
| Vietnam | 6 | Belgium | 6 | South Korea | 6 |
| Portugal | 6 | Morocco | 6 | Finland | 6 |
| Thailand | 6 | South Sudan | 6 | Russia | 6 |
| Switzerland | 6 | Turkey | 6 | Singapore | 6 |

*Table 12.* Distribution of Questions Across Countries.

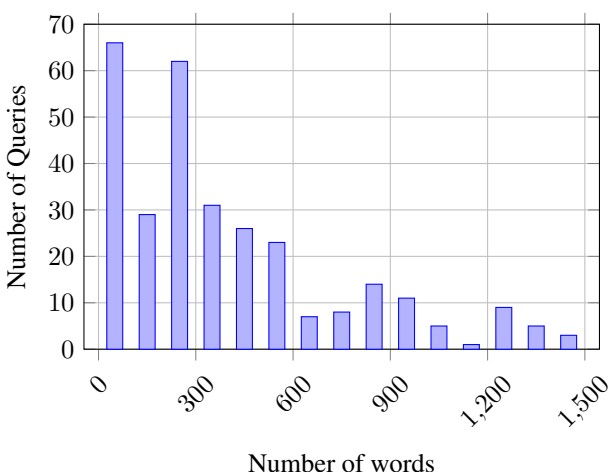

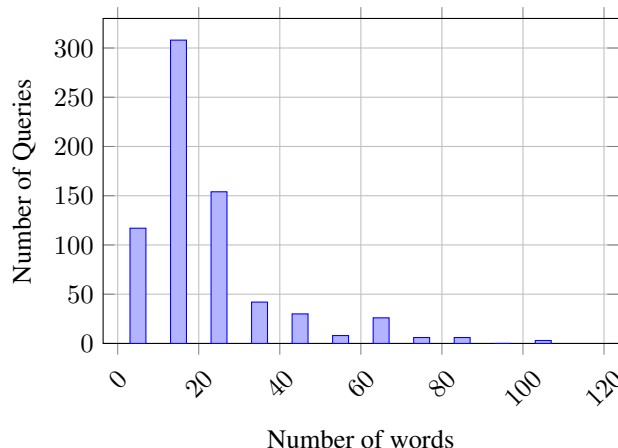

*Figure 5.* Distribution of Textual-Context Lengths (300 samples)

*Figure 6.* Distribution of Question Lengths (700 samples)

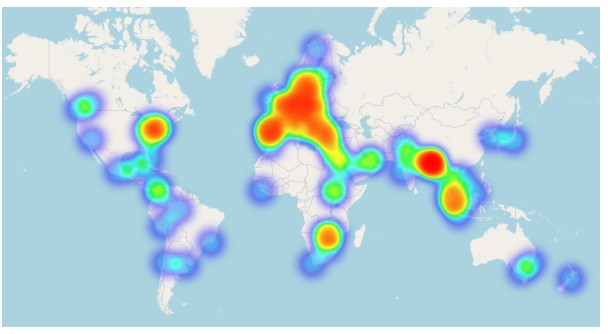

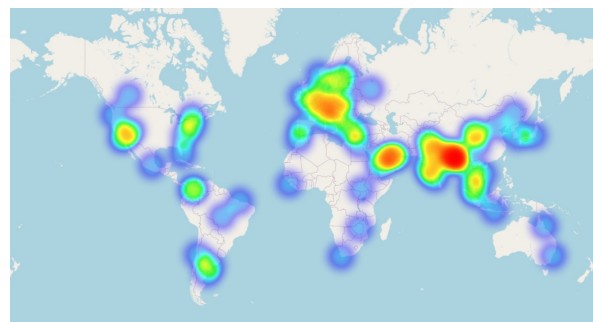

*Figure 6.* (a)

*Figure 6.* (b)

*Figure 7.* Geographical Distribution of Textual and Visual Contexts. The left heatmap (a) represents the locations of places mentioned in textual contexts, while the right heatmap (b) shows the locations derived from map snapshots in visual contexts.

### F.1. Zoom Details

In our dataset, zoom levels range from 8.0 to 21.0, as shown in 10.Each visual context is paired with a Google Maps URL, such as https://www.google.com/maps/@35.7048455,139.763263,16.71z?entry=ttu, where the value before the "z" (e.g., 16.71) represents the zoom level. This allows us to easily extract zoom information directly from the URL, ensuring that each visual context can be accurately mapped to its respective level of detail.

## G. Additional Experiment Results

### G.1. Addressing Failures and Enhancing Geospatial Reasoning in LLMs

For additional experiments, we filtered questions from our textual/API dataset into three subcategories:

1. *Straight-Line Distance* (47 questions): These questions require the computation of straight-line distances, such as "What is the straight-line distance between the Atomium in Brussels and the Belfry of Bruges?"

2. *Cardinal Direction* (24 questions): These involve determining cardinal directions[5], e.g., "What is the direction of the Little Mermaid statue from Copenhagen Central Station?"

3. *Counting* (23 questions): Questions involving counting entities, such as "How many convenience stores are there within a 400 m radius of the Tokyo Tower?"

---

[5]https://en.wikipedia.org/wiki/Cardinal_direction

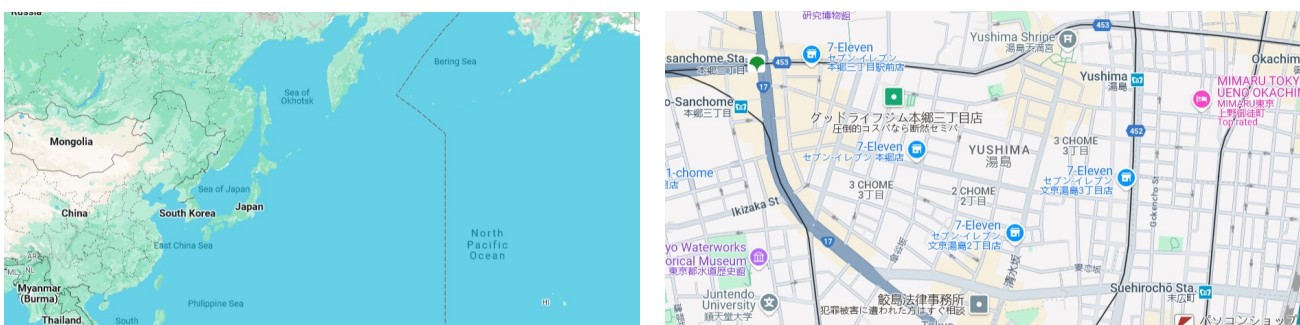

*Figure 8.* Zoom level 1                                      *Figure 9.* Zoom level 16

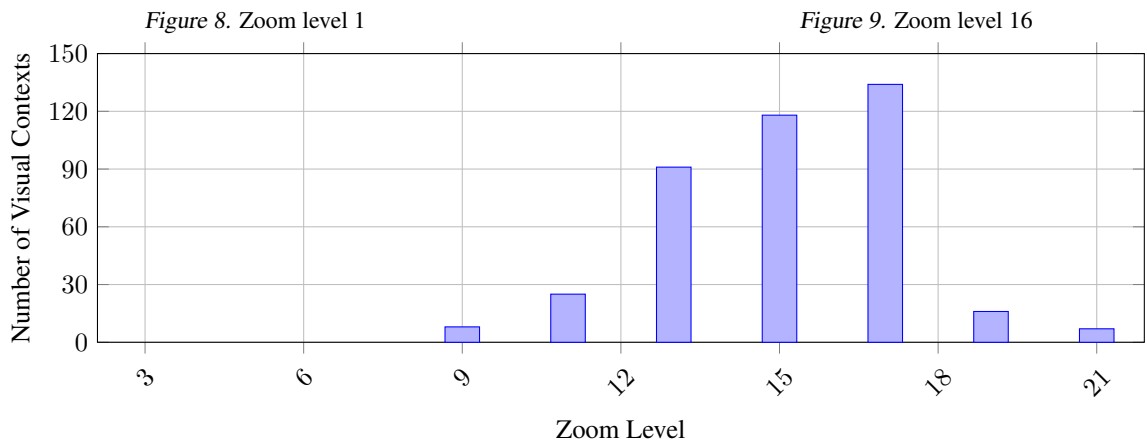

*Figure 10.* Distribution of Zoom Levels (400 Visual Samples)

We visualized the accuracy of various LLMs on these subcategories under the MapEval-Textual setting, with the following findings:

1. *Straight-Line Distance:* Figure 11, illustrates the accuracy on straight-line distance related questions. We can see that all models struggled, with the best accuracy being only 51.06%.

2. *Cardinal Direction:* Figure 12, illustrates the accuracy on cardinal-direction related questions. Here LLMs showed significant variability. While Claude-3.5-Sonnet achieved 91% accuracy, Gemma-2.0-27B scored only 16.67%.

3. *Counting:* Figure 13, illustrates the accuracy on counting related questions. In this case, Claude-3.5-Sonnet underperformed compared to the open-source Gemma-2.0-27B (60.87% accuracy).

We identified a scope for improvement in these areas and enhanced the models' capabilities by integrating external tools (e.g., a calculator) specifically designed for calculating straight-line distances and cardinal directions. For straight-line distances, we employed the Haversine formula[6] to compute the great-circle distance. To determine cardinal directions, we calculated the bearing[7] between two geographic coordinates. Figures 14 and 15 demonstrate a significant improvement in model performance with these tools.

For straight-line distance-related questions, the best accuracy jumped from 51.06% to 85.11%. Similarly, for cardinal-direction questions, the top model achieved an accuracy of 95.83%, compared to the previous maximum of 91.67%. In the case of GPT-4o-mini, these enhancements led to even further progress, with the model demonstrating a leap in both straight-line distance and cardinal direction accuracy, surpassing previous models. In the case of GPT-4o-mini, these enhancements led to even further progress, with the model demonstrating a remarkable leap in both straight-line distance and cardinal direction accuracy. Specifically, the straight-line distance accuracy improved from 34.04% to 78.72%, while cardinal-direction accuracy increased from 29.17% to 91.67%.

---

[6] https://en.wikipedia.org/wiki/Haversine_formula
[7] https://en.wikipedia.org/wiki/Bearing_(navigation)

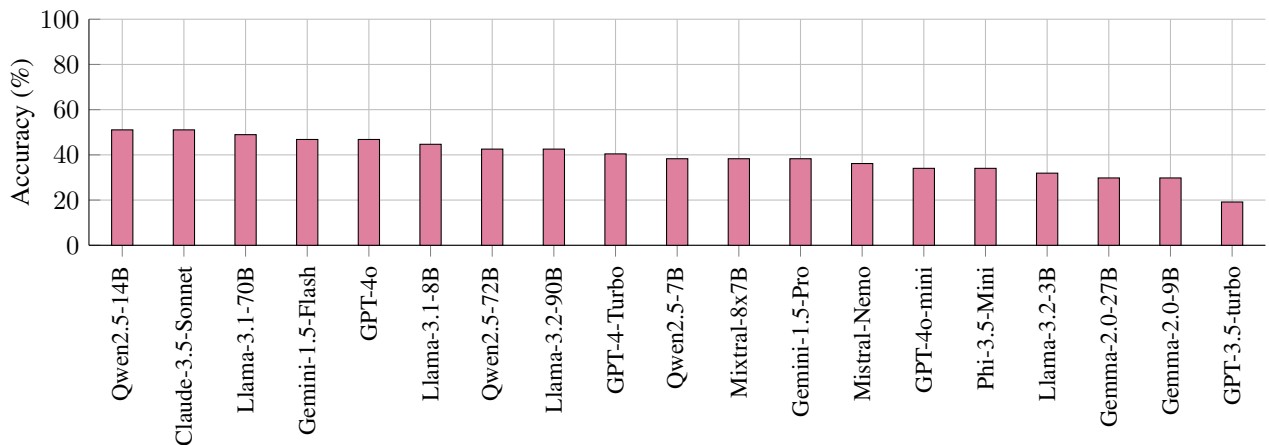

*Figure 11.* Accuracy of LLMs on questions which needs **Straight-Line Distance** computation

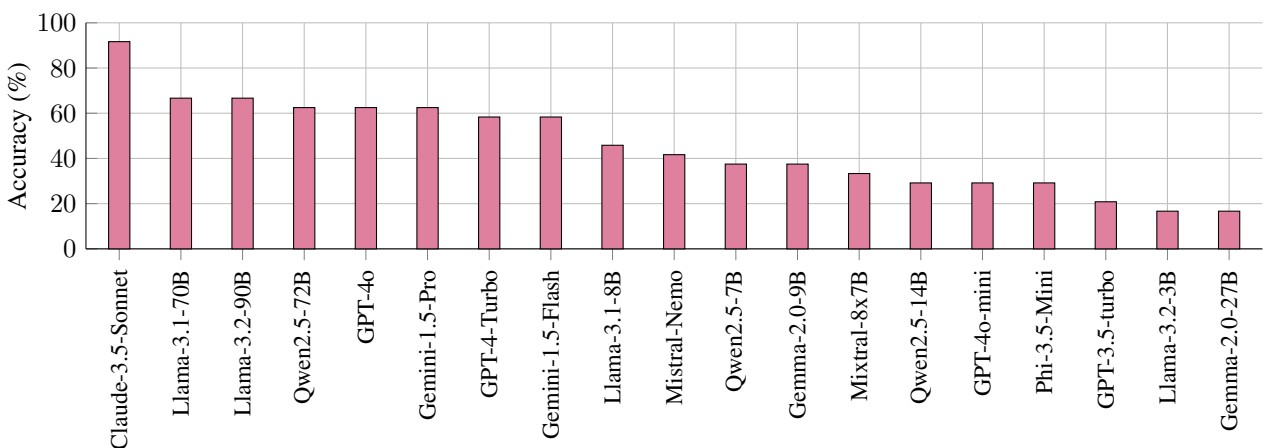

*Figure 12.* Accuracy of LLMs on questions which needs **Cardinal Direction** computation

These results highlight the limitations of current LLMs in handling fine-grained geospatial queries independently and emphasize the value of augmenting LLM capabilities with external computational tools. Future work can explore the integration of more robust external services to address the nuances of spatial reasoning comprehensively.

### G.2. Open-Ended Evaluation

To demonstrate the flexibility of MapEval beyond the MCQ format, we conducted an additional experiment where we removed the answer choices from both textual and visual variants of the benchmark and evaluated open-ended responses from Claude-3.5-Sonnet. The outputs were automatically graded using the o3-mini model.

Tables 13 present the performance results across all categories. While the performance in the open-ended setting is generally lower compared to MCQ, it shows that the queries remain valid and evaluable without answer options. Notably, Claude-3.5-Sonnet still performs competitively in several categories such as *Routing* and *Trip*.

We also observed limitations of automatic evaluation. For example, in the *Unanswerable* category of MapEval-Textual, O3-mini initially assigned an accuracy of 55%, but manual inspection revealed that approximately 80% of Claude-3.5-Sonnet's responses were actually correct. This illustrates the challenge of reliably assessing open-ended spatial reasoning with current tools.

Moreover, open-ended evaluation effectively doubles API costs: each query requires two calls—one to generate a response and another to score it. Since longer, free-form responses use more tokens, cost scales significantly compared to the MCQ

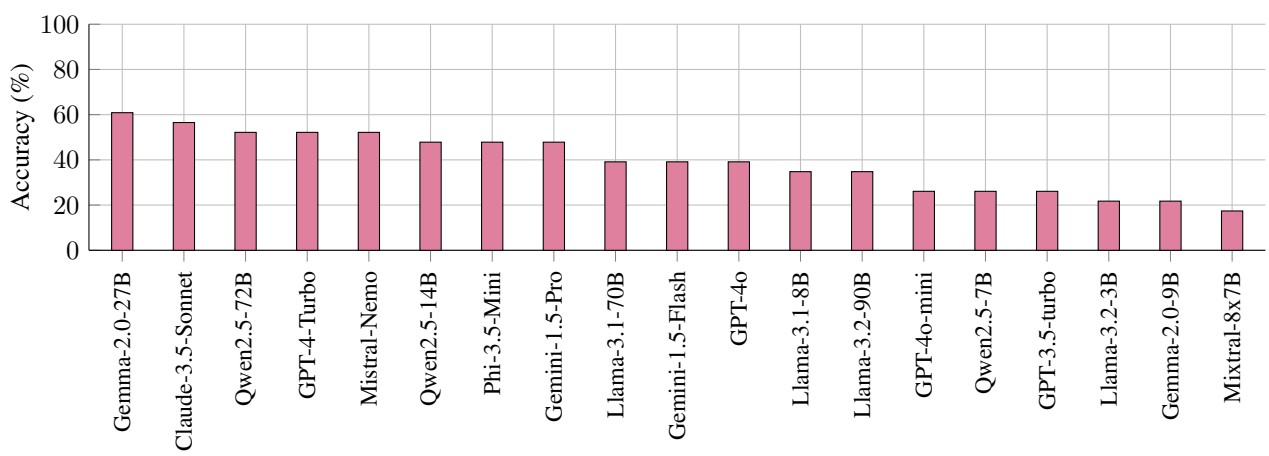

*Figure 13.* Accuracy of LLMs on **Counting** related questions

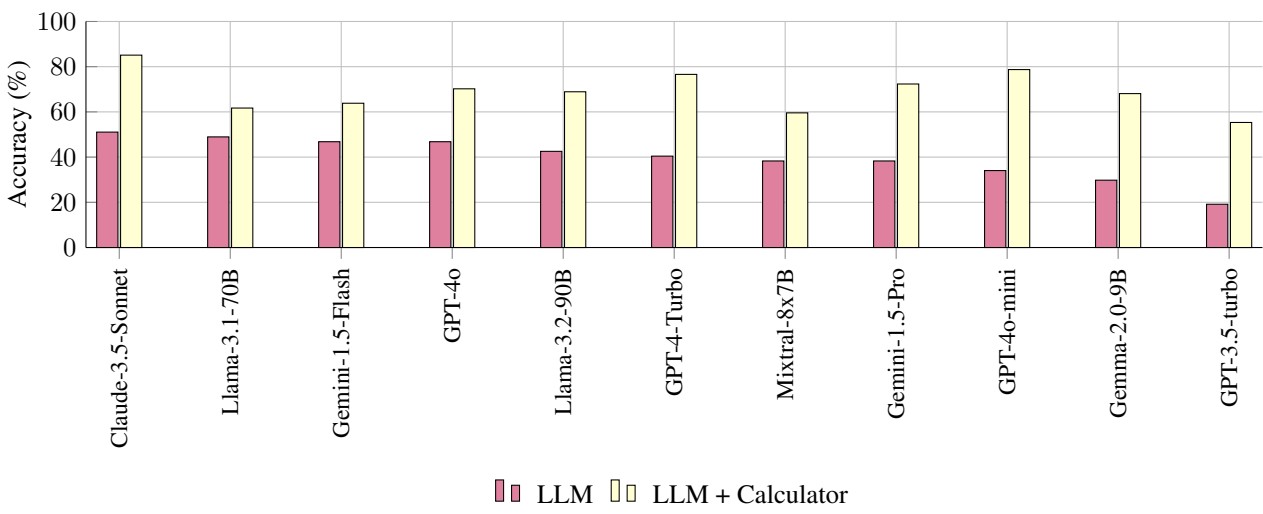

*Figure 14.* Improved accuracy of LLMs after integrating calculator to compute **Straight-Line Distance**

setting.

Thus, while MapEval fully supports open-ended evaluation, our MCQ format remains a more cost-effective and scalable solution for benchmarking LLMs.

| MapEval-Textual | | | | | | |
|---|---|---|---|---|---|---|
| **Evaluation** | **Overall** | **Place Info** | **Nearby** | **Routing** | **Trip** | **Unanswerable** |
| Open-ended | 55.33 | 43.75 | 43.37 | 69.70 | 67.16 | 55.00 |
| MCQ | 66.33 | 73.44 | 73.49 | 75.76 | 49.25 | 40.00 |
| MapEval-Visual | | | | | | |
| **Evaluation** | **Overall** | **Place Info** | **Nearby** | **Routing** | **Counting** | **Unanswerable** |
| Open-ended | 51.88 | 67.70 | 51.11 | 35.00 | 43.18 | 65.00 |
| MCQ | 61.65 | 82.64 | 55.56 | 45.00 | 47.73 | 90.00 |

*Table 13.* Performance of Claude-3.5-Sonnet under open-ended and MCQ settings.

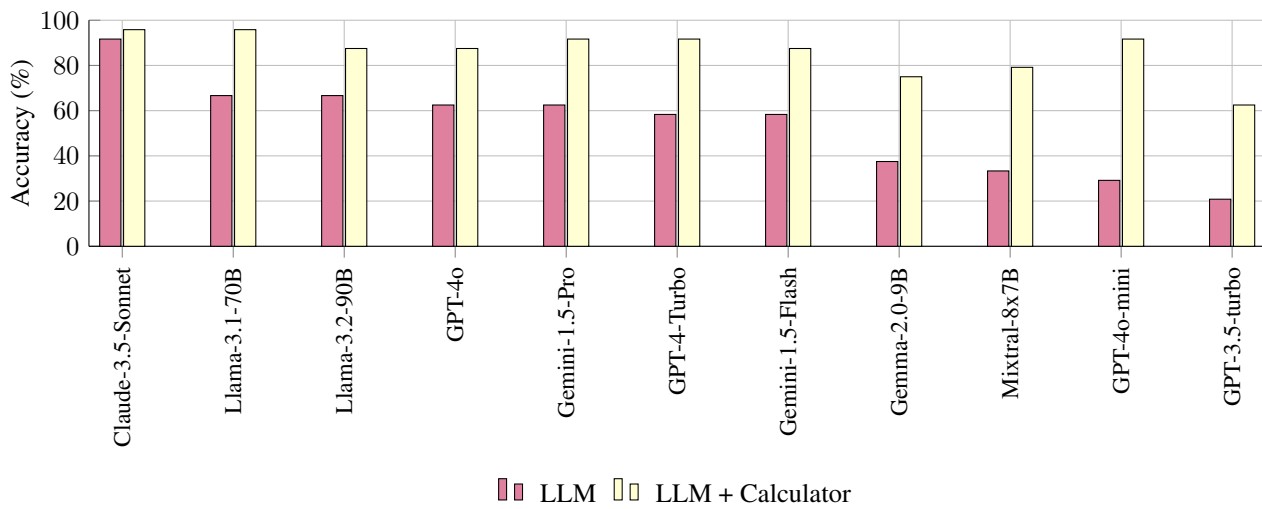

*Figure 15.* Improved accuracy of LLMs after integrating calculator to compute **Cardinal Direction**

### G.3. Fine-Tuning on MapEval

To assess whether the challenges posed by MapEval stem merely from a lack of training exposure, we conducted additional experiments to fine-tune several open-source models on a subset of MapEval-Textual.

We randomly split the 300 MCQ questions into a training set of 97 questions and a test set of 203 questions. We fine-tuned five representative models—Phi-3.5-mini, LLama-3.2-3B, Qwen-2.5-7B, LLama-3.1-8B, and Gemma-2.0-9B—on the training portion using standard instruction tuning.

As shown in Table 14, the performance gains from fine-tuning were generally small ($< 5\%$), and in some cases, performance slightly declined. Even after fine-tuning, all models perform significantly worse than leading LLMs such as GPT-4o, Claude-3.5-Sonnet, or Gemini 1.5 Pro.

These results suggest that the difficulty of MapEval is not solely attributable to a lack of task-specific examples. Instead, the benchmark reveals deeper limitations in current LLMs' ability to handle fine-grained, geospatial reasoning tasks. We believe this highlights the need for more advanced training strategies and dedicated geospatial reasoning capabilities in future LLMs.

| Model | Pretrained | Finetuned |
|---|---|---|
| Phi-3.5-mini | 39.90 | 34.48 |
| LLama-3.2-3B | 34.98 | 35.96 |
| Qwen-2.5-7B | 41.87 | 43.35 |
| LLama-3.1-8B | 46.31 | 44.33 |
| Gemma-2.0-9B | 46.80 | 51.23 |

*Table 14.* Performance comparison of open-source models on MapEval-Textual (test set) before and after fine-tuning on 97 MCQs.

## H. Evaluation Results Visualization

In this section, we present the results of our evaluations through a series of charts that summarize the performance of different models across various categories. These visualizations provide a clear and concise comparison of model effectiveness in addressing textual, API-based, and visual geospatial queries. The charts are designed to highlight key trends, strengths, and limitations of the evaluated approaches.

### H.1. MapEval-Textual

Figure 16 illustrates the performance of models on MapEval-Textual.

Figure 17 illustrates how accuracy of models in MapEval-Textual changes with different context length.

### H.2. MapEval-API

Figure 18 illustrates the performance of models on MapEval-API.

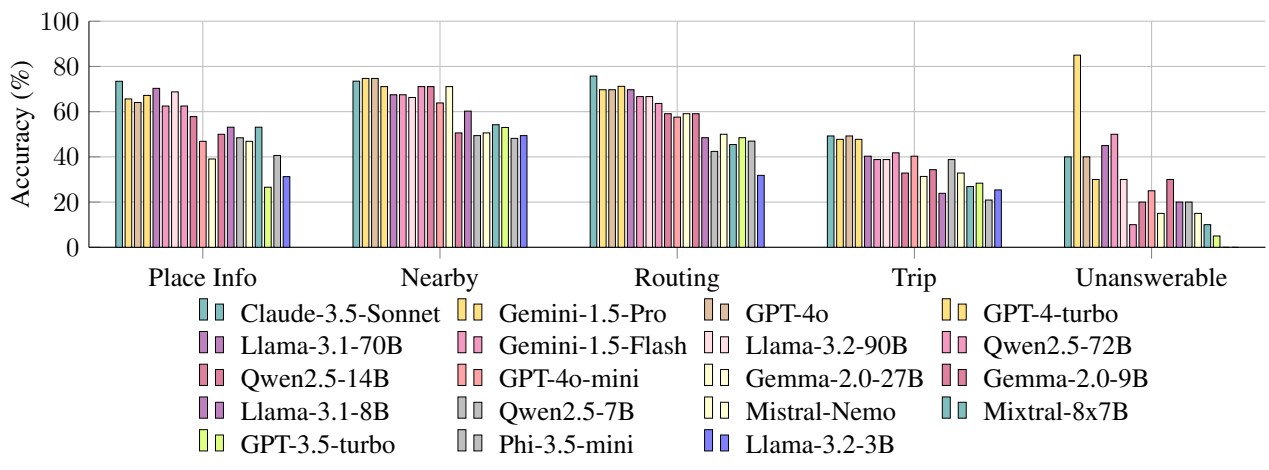

Figure 16. MapEval-Textual categorical accuracy

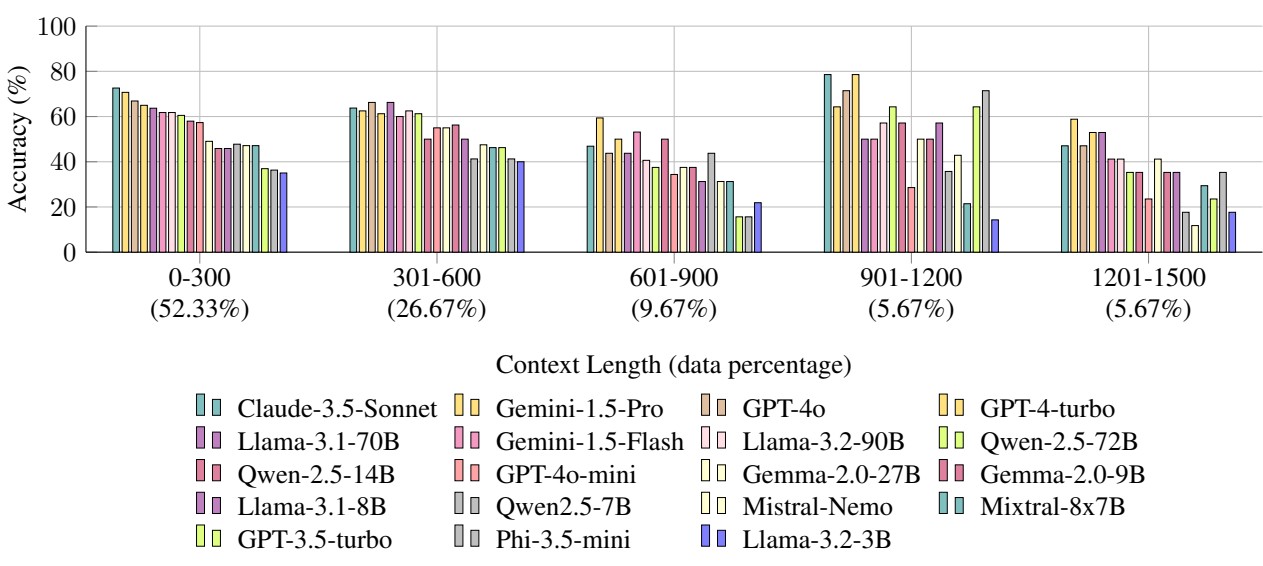

Figure 17. Accuracy vs. Context Length (MapEval-Textual )

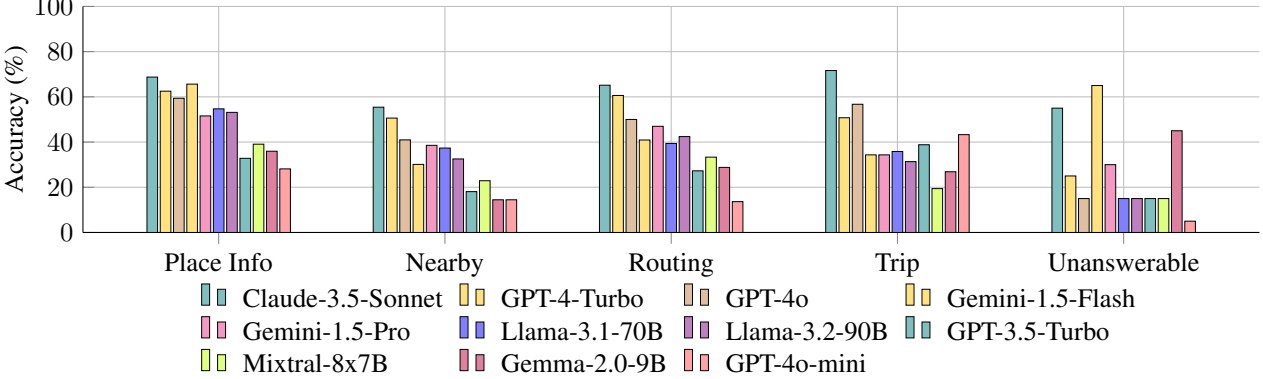

Figure 18. MapEval-API categorical accuracy

Additionally in Figure 19, we can visualize the number of times agent stopped due to iteration limit. This happens when agents repeatedly calls the same api with same parameters and gets the same response. While GPT-3.5-Turbo encounters 16 infinite iterations, Claude-3.5-Sonnet doesn't face this issue.

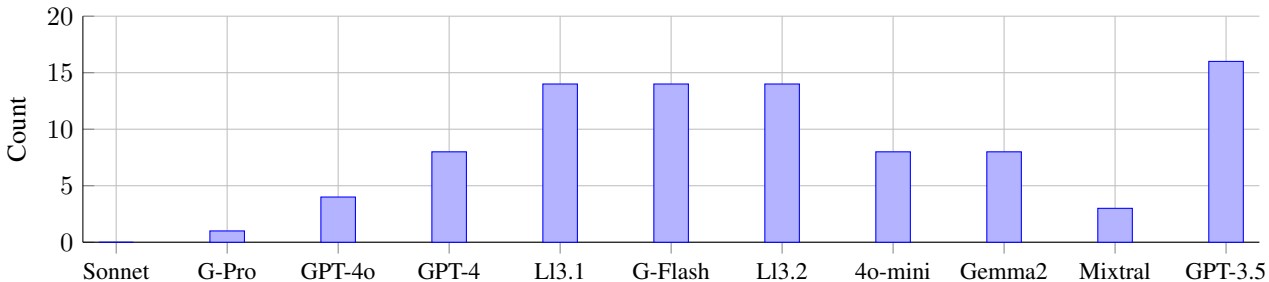

Figure 19. Number of times agent stopped due to iteration limit

### H.3. MapEval-Visual

Figure 20 illustrates the performance of models on MapEval-Visual.

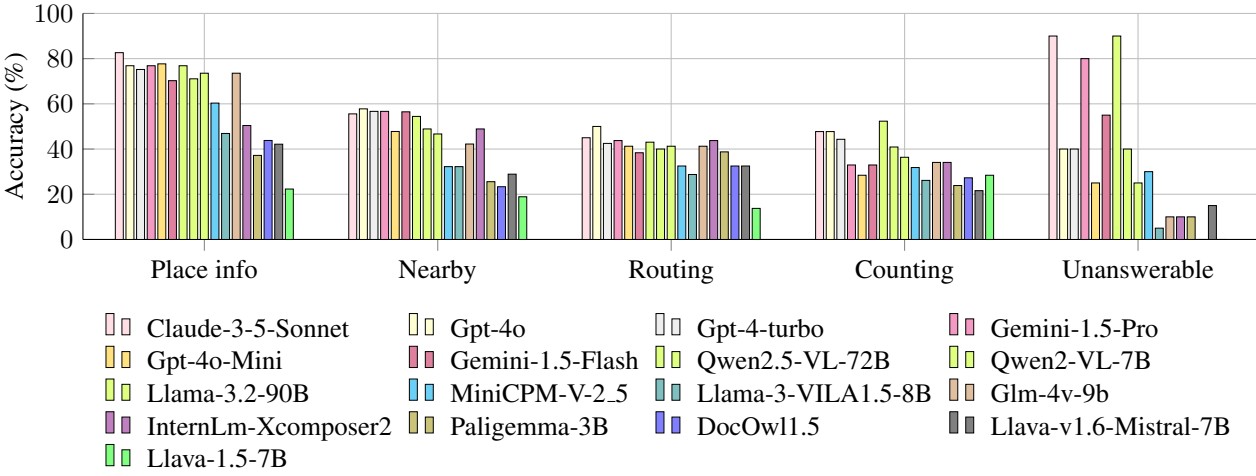

Figure 20. MapEval-Visual categorical accuracy

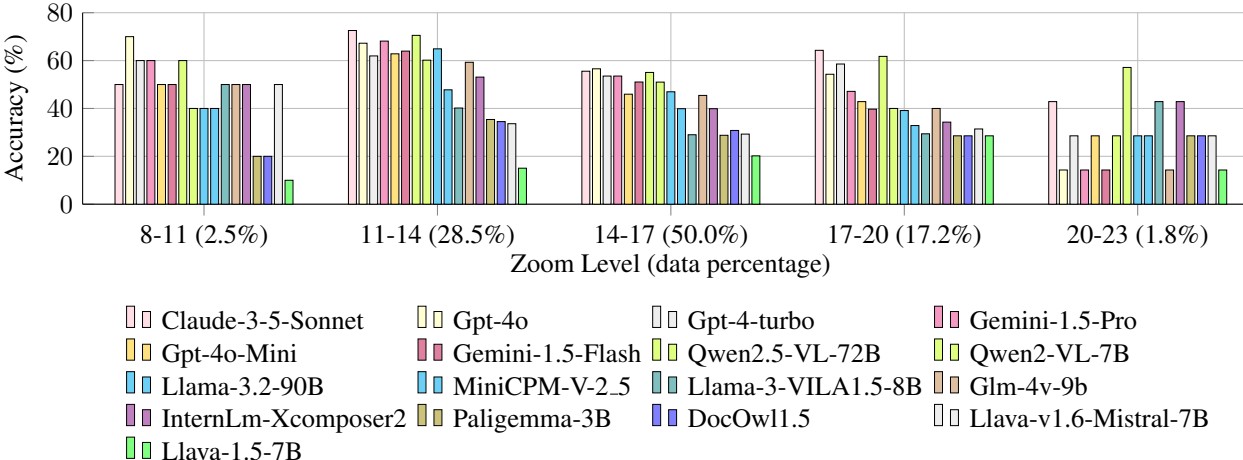

Figure 21. Accuracy by Zoom Level.

# I. Qualitative Examples

| Type | Task | Question Example |
|---|---|---|
| Place Info | Textual/API | Which coffee shop is situated between Louvre Museum and Eiffel Tower? |
| | | What is the direction and straight-line distance from Victoria Falls to Hwange National Park? |
| | Visual | I'm at Baridhara K Block, feeling unwell, and need some medicine. What is a nearby pharmacy with a good rating that is open? |
| Nearby | Textual/API | How many shopping malls are there within a 500 m radius of Berlin Cathedral? |
| | | I am at Toronto Zoo. Today is Sunday and it's currently 8:30 PM. How many nearby ATMs are open now? |
| | Visual | I'm currently staying at Hörselberg-Hainich, while my friend is staying at Tüngeda. After we meet up, I want to visit an amusement park nearby. Can you suggest one that's close to us? |
| Routing | Textual/API | I want to walk from D03 Flame Tree Ridge to Aster Cedars Hospital, Jebel Ali. Which walking route involves taking the pedestrian overpass? |
| | | On the driving route from Hassan II Mosque to Koutoubia via A3, how many roundabouts I will encounter in total? |
| | Visual | Which restaurant is on the left side of the route from Metro El Golf to Metro Tobalaca L1? |
| Unanswerable | Textual/API | How many food stalls are there north of the overbridge at Hakaniemi? |
| | | Find a good coffee shop on the left side of my driving path from my home near Petaling Jaya to my office in Kuala Lumpur. |
| | Visual | How much time it would take to go to Igreja Nossa Senhora Da Conceição do Coroadinho - Matriz? |
| Trip | Textual/API | I live in Indira Road. At tomorrow 2 pm I will leave my house. I need to go to Military Museum to visit with friends for 2 hours and Multiplan Center to buy a keyboard (which will take 20 minutes) and Sonali Bank, BUET to receive my check book (which will take 30 minutes). In which order I should visit the places so that I reach there on time and come back home as early as possible. I will use public transport. |
| Counting | Visual | How many restaurants or clubs are on the bottom side of Linnakatu road? |

*Table 15.* Additional Complex Examples

*Listing 1.* Example evaluation of MapEval-Textual `Green` : Correct Answer. `Red` : Wrong Answer.

`Context:`
Information of Eiffel Tower:
- Location: Av. Gustave Eiffel, 75007 Paris, France(48.8584, 2.2945).

Information of Mont Saint-Michel:
- Location: 50170 Mont Saint-Michel, France(48.6361, -1.5115).

─────

`Query:`
What is the straight-line distance between the Eiffel Tower in Paris, France, and the Mont Saint-Michel in Normandy, France?

─────

`Prompt:`
Please respond in the following JSON format:
{
  "option_no": <option index>, // "option_no" refers to the number corresponding to the chosen answer from the list
  of options. It should be between 1 and 4.
  "explanation": "<reason>"
}

Example Prompt:
Question: What is the capital of France?
Option1: Berlin
Option2: Paris
Option3: Madrid
Option4: Rome

Example Response:
{
  "option_no": 2,
  "explanation": "Paris is the capital of France."
}

Provide your answer in this format. Remember to answer the question strictly based on the given context, without
using any external knowledge or assumptions.

─────

`Options:`
1. 230 km
2. 250 km
`3.`  280 km
4. 310 km

─────

`Claude-3.5-Sonnet:`
To answer this question, I'll need to calculate the straight-line distance between the two points using their given
coordinates. I'll use the Haversine formula for this calculation. Eiffel Tower: (48.8584, 2.2945) Mont Saint-Michel:
 (48.6361, -1.5115) After performing the calculation, the approximate distance is 284 km. Based on this result, the
closest option is 280 km.
`Gemini-1.5-Pro:`
I cannot calculate the straight-line distance between the Eiffel Tower and Mont Saint-Michel without using external
knowledge or tools. This calculation requires geographic data and mathematical formulas.
`GPT-4o:`
The straight-line distance between the coordinates of the Eiffel Tower (48.8584, 2.2945) and Mont Saint-Michel
(48.6361, -1.5115) is approximately 280 km.
`Gemma-2.0-9B:`
To determine the straight-line distance, we'd need to use the coordinates provided and apply a distance formula.
Unfortunately, I can't perform calculations. **Therefore, I cannot answer the question based on the given context.**

*Listing 2.* Example evaluation of MapEval-Textual `Green` : Correct Answer. `Red` : Wrong Answer.

`Context:`
```
Information of Seurasaari Open-Air Museum:
- Location: Seurasaari, 00250 Helsinki, Finland(60.1856, 24.8846).

Information of Helsinki Central Station:
- Location: Kaivokatu 1, 00100 Helsinki, Finland(60.1719, 24.9414).
```

`Query:`
```
What is the direction of the Seurasaari Open-Air Museum from Helsinki Central Station?
```

`Prompt:`
```
Please respond in the following JSON format:
{
  "option_no": <option index>, // "option_no" refers to the number corresponding to the chosen answer from the list
  of options. It should be between 1 and 4.
  "explanation": "<reason>"
}

Example Prompt:
Question: What is the capital of France?
Option1: Berlin
Option2: Paris
Option3: Madrid
Option4: Rome

Example Response:
{
  "option_no": 2,
  "explanation": "Paris is the capital of France."
}

Provide your answer in this format. Remember to answer the question strictly based on the given context, without
using any external knowledge or assumptions.
```

`Options:`
```
1. East
```
`2.` West
```
3. North
4. South
```

`Calude-3.5-Sonnet:`
```
To determine the direction of Seurasaari Open-Air Museum from Helsinki Central Station, I'll compare their
coordinates: Seurasaari Open-Air Museum: 60.1856, 24.8846 Helsinki Central Station: 60.1719, 24.9414 Analyzing the
coordinates: 1. Latitude: Seurasaari (60.1856) is slightly north of Helsinki Central Station (60.1719) 2. Longitude:
 Seurasaari (24.8846) is west of Helsinki Central Station (24.9414) The primary direction is west, with a slight
northward component.
```
`Gemini-1.5-Pro:`
```
Seurasaari Open-Air Museum (60.1856, 24.8846) is located at a higher latitude and almost same longitude compared to
Helsinki Central Station (60.1719, 24.9414). Higher latitude corresponds to North.
```
`GPT-4o:`
```
Seurasaari Open-Air Museum is located to the north of Helsinki Central Station based on their coordinates.
```
`Gemma-2.0-9B:`
```
Seurasaari Open-Air Museum is located at 60.1856, 24.8846 and Helsinki Central Station is located at 60.1719,
24.9414. The museum's coordinates are further north.
```

*Listing 3.* Example evaluation of MapEval-Textual `Green` : Correct Answer. `Red` : Wrong Answer.

`Context:`
Nearby Restaurants of St. Lawrence Market are:
1. A&W Canada
- Open: `Monday: Open 24 hours` , Tuesday: Open 24 hours, Wednesday: Open 24 hours, Thursday: Open 24 hours, Friday: Open 24 hours, Saturday: Open 24 hours, Sunday: Open 24 hours.
....
7. Quik Sushi
- Open: `Monday: 9:00AM-7:00PM` , Tuesday: 11:30AM-7:30PM, Wednesday: 9:00AM-7:00PM, Thursday: 9:00AM-7:00PM, Friday: 9:00AM-7:00PM, Saturday: 9:00AM-7:00PM, Sunday: 9:00AM-7:00PM.
....
9. Big Pita
- Open: `Monday: 10:30AM-9:00PM` , Tuesday: 10:30AM-9:00PM, Wednesday: 10:30AM-9:00PM, Thursday: 10:30AM-9:00PM, Friday: 10:30AM-9:00PM, Saturday: 10:30AM-9:00PM, Sunday: 10:30AM-9:00PM.
....
14. Pizza Pizza
- Open: `Monday: 11:00AM-1:00AM` , Tuesday: 11:00AM-1:00AM, Wednesday: 11:00AM-1:00AM, Thursday: 11:00AM-1:00AM, Friday: 11:00AM-2:00AM, Saturday: 11:00AM-2:00AM, Sunday: 11:00AM-1:00AM.
....
17. Market Street Catch
- Open: `Monday: 11:00AM-8:00PM` , Tuesday: 11:00AM-9:00PM, Wednesday: 11:00AM-9:00PM, Thursday: 11:00AM-9:00PM, Friday: 11:00AM-10:00PM, Saturday: 11:00AM-10:00PM, Sunday: 11:00AM-8:00PM.
....
20. Churrasco Of The St Lawrence
- Open: Monday: Closed, Tuesday: 7:00AM-7:00PM, Wednesday: 7:00AM-7:00PM, Thursday: 7:00AM-7:00PM, Friday: 7:00AM-7:00PM, Saturday: 7:00AM-7:00PM, Sunday: 10:00AM-5:00PM.

`Query:`
How many nearby restaurants of St. Lawrence Market are open on Monday?

`Prompt:`
Please respond in the following JSON format:
{
  "option_no": <option index>, // "option_no" refers to the number corresponding to the chosen answer from the list
  of options. It should be between 1 and 4.
  "explanation": "<reason>"
}
Example Prompt:
Question: What is the capital of France?
Option1: Berlin, Option2: Paris, Option3: Madrid, Option4: Rome
Example Response:
{
  "option_no": 2,
  "explanation": "Paris is the capital of France."
}
Provide your answer in this format. Remember to answer the question strictly based on the given context, without using any external knowledge or assumptions.

`Options:`
1. 2
2. 3
3. 4
`4.` 5

`Calude-3.5-Sonnet:`
Based on the given information, 3 nearby restaurants are open on Monday. These are: A&W Canada (open 24 hours), Quik Sushi (open 9:00 AM - 7:00 PM), and Big Pita (open 10:30 AM - 9:00 PM). All other listed restaurants are either closed on Monday or their Monday hours are not specified in the provided information.
`Gemini-1.5-Pro:`
Based on the provided information, these restaurants are open on Monday: A&W Canada, Quik Sushi, Pizza Pizza, and Market Street Catch.
`GPT-4o:`
Only A&W Canada and Quik Sushi are open on Monday.
`Gemma-2.0-9B:`
Only A&W Canada is open 24 hours on Monday.

*Listing 4.* Example evaluation of MapEval-Textual. `Green` : Correct Answer. `Red` : Wrong Answer.

`Context:`

```
Information of The Metropolitan Museum of Art:
- Location: 1000 5th Ave, New York, NY 10028, USA.
- Open: Monday: 10:00AM-5:00PM, Tuesday: 10:00AM-5:00PM, Wednesday: Closed, Thursday: 10:00AM-5:00PM, Friday: 10:00
AM-9:00PM, Saturday: 10:00AM-9:00-PM, Sunday: 10:00AM-5:00PM.

....

Travel Time from The Metropolitan Museum of Art to American Wing Cafe on foot is 5 mins (0.3 km).

Travel Time from The Metropolitan Museum of Art to The Great Hall Balcony Cafe on foot is 2 mins (0.1 km).

....

Nearby Cafes of The Metropolitan Museum of Art are (sorted by distance in ascending order):
1. American Wing Cafe (1000 5th Avenue, New York)
2. The Great Hall Balcony Cafe (1000 5th Avenue 2nd Floor, New York)
3. Petrie Court Cafe (1000 5th Avenue, New York)
4. Starbucks (1142 Madison Avenue, New York)
.....
```

`Query:`

```
Today is Monday. I have an afternoon free in New York and plan to visit The Metropolitan Museum of Art for 3 hours,
followed by a 30-minute coffee break at a nearby cafe, and then spend 1 hour in Central Park (New York, NY, USA).
Plan a schedule to ensure I have enough time for everything. I will walk between places.
```

`Prompt:`

```
Please respond in the following JSON format:
{
  "option_no": <option index>, // "option_no" refers to the number corresponding to the chosen answer from the list
  of options. It should be between 1 and 4.
  "explanation": "<reason>"
}
Example Prompt:
Question: What is the capital of France?
Option1: Berlin, Option2: Paris, Option3: Madrid, Option4: Rome
Example Response:
{
  "option_no": 2,
  "explanation": "Paris is the capital of France."
}
Provide your answer in this format. Remember to answer the question strictly based on the given context, without
using any external knowledge or assumptions.
```

`Options:`

```
1. (3.00PM - 6.00PM) Visit The Metropolitan Museum of Art → (6.02PM - 6.32PM) Have coffee at The Great Hall Balcony
 Cafe → (6.38PM - 7.38PM) Spend time in Central Park.
2. (3.00PM - 6.00PM) Visit The Metropolitan Museum of Art → (6.08PM - 6.38PM) Have coffee at  Central Park cafe →
(6.42PM - 7.42PM) Spend time in Central Park.
3. (3.00PM - 6.00PM) Visit The Metropolitan Museum of Art → (6.05PM - 6.35PM) Have coffee at Starbucks  → (6.39
PM - 7.39PM) Spend time in Central Park.
4. (2.00PM - 5.00PM) Visit The Metropolitan Museum of Art → (5.09PM - 5.39PM) Have coffee at Matto Espresso →
(5.47PM - 6.47PM) Spend time in Central Park.
```

`Calude-3.5-Sonnet:`

Option 3 is the most suitable schedule based on the given information. The Metropolitan Museum of Art is open on
Mondays from 10:00 AM to 5:00 PM, so visiting from 3:00 PM to 6:00 PM is possible. Starbucks is open until 7:30 PM
on Mondays, making it a viable option for a coffee break after the museum visit. Central Park is open until 1:00 AM,
 so spending time there from 6:39 PM to 7:39 PM is feasible.

`Gemini-1.5-Pro:`

Option3 is the only option that has time slots within the opening hours of all the places mentioned. The
Metropolitan Museum of Art is open until 5:00PM on Monday, Starbucks is open until 7:30PM and Central Park is open
until 1:00AM.

`GPT-4o:`

Option 3 is feasible as Starbucks is open on Monday and is nearby. The schedule allows for a 3-hour visit to The
Metropolitan Museum of Art, a 30-minute coffee break, and 1 hour in Central Park.

`Gemma-2.0-9B:`

Option 1 provides the most feasible schedule. It allows for a 3-hour visit to the museum, a 30-minute coffee break
at The Great Hall Balcony Cafe (which is within walking distance), and a 1-hour visit to Central Park, all while
ensuring enough time for travel between locations.

*Listing 5.* Example evaluation of MapEval-Textual `Green` : Correct Answer. `Red` : Wrong Answer.

`Context:`
There are 3 routes from Petaling Jaya to Federal Territory of Kuala Lumpur by car. They are:
1. Via Damansara Link/Lebuhraya SPRINT/Sistem Penyuraian Trafik Kuala Lumpur Barat/E23 | 21 mins | 15.2 km
- Head east on Persiaran Tropicana toward Persiaran Tropicana
....
3. Via Damansara Link/Lebuhraya SPRINT/Sistem Penyuraian Trafik Kuala Lumpur Barat/E23 and Jalan Maarof | 25 mins | 15.2 km
- Head east on Persiaran Tropicana toward Persiaran Tropicana
....
Nearby Cafes of Petaling Jaya are (sorted by distance in ascending order):
1. Dareia Coffee
- Address: Ps2-13 Lumi Tropicana, Persiaran Tropicana, Tropicana, Petaling Jaya.
- Rating: 5. (7 ratings).
....
Nearby Cafes of Federal Territory of Kuala Lumpur are (sorted by distance in ascending order):
1. Gwaku Coffee
- Address: ULG Lobby Menara TH 1 Sentral, Jalan Stesen Sentral 5, Kuala Lumpur Sentral, Kuala Lumpur.
- Rating: 4.2. (62 ratings).
....

`Query:`
Find a good coffee shop on the left side of my driving path from my home near Petaling Jaya to my office in Kuala Lumpur.

`Prompt:`
Please respond in the following JSON format:
{
  "option_no": <option index>, // "option_no" refers to the number corresponding to the chosen answer from the list
  of options. It should be between 0 and 4.
  "explanation": "<reason>"
}

Example Prompt:
Question: What is the capital of France?
Option0: Unanswerable, Option1: Berlin, Option2: Paris, Option3: Madrid, Option4: Rome
Example Response:
{
  "option_no": 2,
  "explanation": "Paris is the capital of France."
}
Provide your answer in this format. Remember to answer the question strictly based on the given context, without using any external knowledge or assumptions.

`Options:`
`0.` Unanswerable
1. Dareia Coffee
2. Gwaku Coffee
3. The Lemon Tree
4. Coffeeloca

`Calude-3.5-Sonnet:`
The Lemon Tree is a good coffee shop option on the left side of the driving path from Petaling Jaya to Kuala Lumpur. It is located at the Ground Floor of Menara Lien Hoe, which is along Persiaran Tropicana. This cafe is on the initial part of the journey and has a rating of 4 out of 5 based on 836 ratings, indicating it's a popular and well-liked spot. It's also described as having a 'Moderate' price level, making it a suitable choice for a coffee stop on the way to the office.
`Gemini-1.5-Pro:`
There is no information on which side of the driving path the cafes are located.
`GPT-4o:`
Gwaku Coffee is located on Jalan Stesen Sentral 5, which is on the driving path from Petaling Jaya to Kuala Lumpur.
`Gemma-2.0-9B:`
Gwaku Coffee is located in Kuala Lumpur Sentral, which is on the driving path from Petaling Jaya to Kuala Lumpur.

*Listing 6.* MapEval-API System Prompt

```
Respond to the human as helpfully and accurately as possible. You have access to the following tools:

PlaceSearch: Get place ID for a given location., args: {'placeName': {'description': 'Name of the place', 'title': '
Placename', 'type': 'string', 'placeAddress': {'description': 'Address of the place', 'title': 'Placeaddress', 'type
': 'string'}}}

PlaceDetails: Get details for a given place ID., args: {'placeId': {'description': 'Place Id of the location', '
title': 'Placeid', 'type': 'string'}}

NearbySearch: Get nearby places around a location., args: {'placeId': {'description': 'The id of the place around
which to retrieve nearby places.', 'title': 'Placeid', 'type': 'string'}, 'type': {'description': 'Type of place (e.
g., restaurant, hospital, etc). Restricts the results to places matching the specified type.', 'title': 'Type', '
type': 'string'}, 'rankby': {'default': 'distance', 'description': 'Specifies the order in which places are listed.
Possible values are: (1. prominence (default): This option sorts results based on their importance. When prominence
is specified, the radius parameter is required. 2. distance: This option sorts places in ascending order by their
distance from the specified location. When distance is specified, radius is disallowed. In case you are not
concerned about the radius, use rankby as distance.)', 'title': 'Rankby', 'type': 'string'}, 'radius': {'anyOf': [{'
type': 'integer'}, {'type': 'null'}], 'default': None, 'description': 'Defines the distance (in meters) within which
 to return place results.', 'title': 'Radius'}}

TravelTime: Estimate the travel time between two places., args: {'originId': {'description': 'Place Id of Origin', '
title': 'Originid', 'type': 'string'}, 'destinationId': {'description': 'Place Id of Destination', 'title': '
Destinationid', 'type': 'string'}, 'travelMode': {'description': 'Mode of transportation (driving, walking,
bicycling, transit)', 'title': 'Travelmode', 'type': 'string'}}

Directions: Get directions/routes between two places., args: {'originId': {'description': 'Place Id of Origin', '
title': 'Originid', 'type': 'string'}, 'destinationId': {'description': 'Place Id of Destination', 'title': '
Destinationid', 'type': 'string'}, 'travelMode': {'description': 'Mode of transportation (driving, walking,
bicycling, transit)', 'title': 'Travelmode', 'type': 'string'}}

Use a json blob to specify a tool by providing an action key (tool name) and an action_input key (tool input).

Valid "action" values: "Final Answer" or PlaceSearch, PlaceDetails, NearbySearch, TravelTime, Directions

Provide only ONE action per $JSON_BLOB, as shown:

```
{
  "action": $TOOL_NAME,
  "action_input": $INPUT
}
```

Follow this format:

Question: input question to answer
Thought: consider previous and subsequent steps
Action:
```
$JSON_BLOB
```
Observation: action result
... (repeat Thought/Action/Observation N times)
Thought: I know what to respond
Action:
```
{
  "action": "Final Answer",
  "action_input": "Final response to human"
}
```

Begin! Reminder to ALWAYS respond with a valid json blob of a single action. Use tools if necessary. Respond
directly if appropriate. Format is Action:```$JSON_BLOB```then Observation:.
Thought:
```

*Listing 7.* Example evaluation of MapEval-API. Green : Correct Answer. Red : Wrong Answer.

Query:
Can I visit The Great Pyramid of Giza at 5:00 PM Saturday?

Options:
1. Yes
2. No

Calude-3.5-Sonnet:

To answer this question accurately, I'll need to check the opening hours of The Great Pyramid of Giza. Let me start by getting the place ID for the location.

Action:
```
{
  "action": "PlaceSearch",
  "action_input": {
    "placeName": "The Great Pyramid of Giza",
    "placeAddress": "Al Haram, Nazlet El-Semman, Al Giza Desert, Giza Governorate, Egypt"
  }
}
```

Observation: ChIJGymPrIdFWBQRJCSloj8vDIE
Thought:Now that I have the place ID for The Great Pyramid of Giza, I'll use it to get more details about the location, including its opening hours.

Action:
```
{
  "action": "PlaceDetails",
  "action_input": {
    "placeId": "ChIJGymPrIdFWBQRJCSloj8vDIE"
  }
}
```

Observation:
- Location: Al Haram, Nazlet El-Semman, Al Haram, Giza Governorate 3512201, Egypt (29.9791705, 31.1342046).
- Open: Monday: 8:00AM-4:30PM, Tuesday: 8:00AM-4:30PM, Wednesday: 8:00AM-4:30PM, Thursday: 8:00AM-4:30PM, Friday: 8:00AM-4:30PM, Saturday: 8:00AM-4:30PM, Sunday: 8:00AM-4:30PM.
- Rating: 4.6. (27048 ratings).

Thought:Based on the information provided by the PlaceDetails tool, I can now answer the question.

The Great Pyramid of Giza is open on Saturdays from 8:00 AM to 4:30 PM. The question asks if it can be visited at 5:00 PM on Saturday. Since 5:00 PM is after the closing time of 4:30 PM, the answer is No.

Therefore, the correct option is Option 2: No.

Action:
```
{
  "action": "Final Answer",
  "action_input": "2"
}
```

*Listing 8.* Example evaluation of MapEval-API `Green` : Correct Answer. `Red` : Wrong Answer.

`Query:`
I am currently staying at Khaosan Tokyo Origami in Asakusa, Tokyo. I want to visit the Senso-ji Temple for 1 hour, Shibuya Crossing for 1.5 hours, Ueno Park for 2 hours, and the Tokyo Skytree for 1.5 hours. I have 3.5 hours available. I will leave my hostel at 7 am. How many places can I visit? I will walk between places.
————
`Options:`
1. One
`2.` Two
3. Three
4. Four
————
Calude-3.5-Sonnet:
To answer this question accurately, I'll need to calculate the travel times between these locations and the time spent at each place. Let's break this down step by step.

First, I'll need to get the place IDs for all the locations mentioned.

Action:
```
{
  "action": "PlaceSearch",
  "action_input": {
    "placeName": "Khaosan Tokyo Origami",
    "placeAddress": "Asakusa, Tokyo"
  }
}
```

.....

Observation: ChIJ35ov0dCOGGARKvdDH7NPHX0
Thought:Thank you for providing the place ID for Tokyo Skytree. Now that we have all the place IDs, let's calculate the travel times between these locations using the walking mode of transportation.

Let's start with the travel time from Khaosan Tokyo Origami to Senso-ji Temple.

Action:
```
{
  "action": "TravelTime",
  "action_input": {
    "originId": "ChIJ4ehN78GOGGARyvT_Bjj5s-c",
    "destinationId": "ChIJ8T1GpMGOGGARDYGSgpooDWw",
    "travelMode": "walking"
  }
}
```

Response:  [[{'distance': {'text': '0.3 km', 'value': 267}, 'duration': {'text': '4 mins', 'value': 227}, 'status': 'LOCAL'}]]

.....

Observation: Travel Time on foot is 51 mins (3.5 km).
Thought:Now that we have all the travel times and the time to be spent at each location, let's calculate how many places can be visited within the 3.5 hours available, starting at 7 am.

1. Khaosan Tokyo Origami to Senso-ji Temple: 4 minutes
2. Time at Senso-ji Temple: 1 hour
3. Total time so far: 1 hour 4 minutes

At this point, it's 8:04 am, and we've used 1 hour 4 minutes of the available 3.5 hours.

The next destination, Shibuya Crossing, is 2 hours 52 minutes away on foot, which would exceed the remaining time (2 hours 26 minutes) even before spending any time there.

Therefore, the traveler can only visit one place (Senso-ji Temple) within the given time constraints.

`Action:`
```
{
  "action": "Final Answer",
  "action_input": "Option1"
}
```

*Listing 9.* Example evaluation of MapEval-Visual Green : Correct Answer. Red : Wrong Answer.

`Context:`

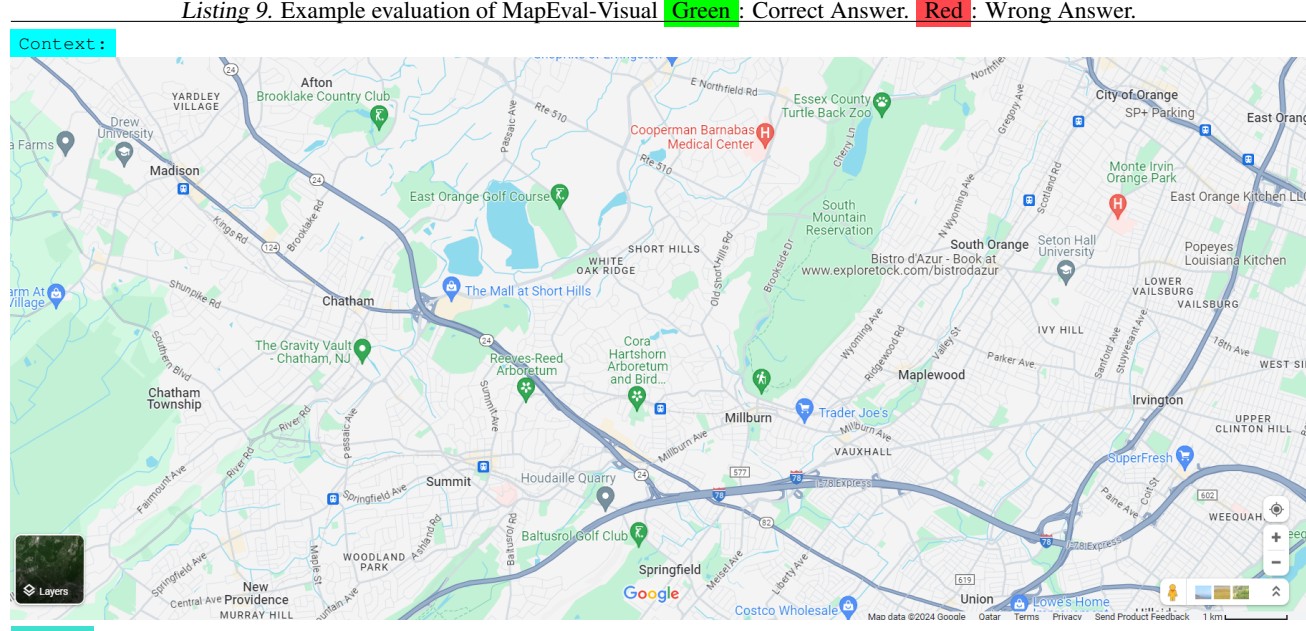

`Query:`
Which Golf Club is located at the intersection of Springfield  and Houdaille Quarry?

`Prompt:`
Choose the correct option from the image also show the answer with visible quotation and give proper explanation.

`Options:`
1.Brooklake Country Club
2. Baltusrol Golf Club
3.East Orange Golf Course
4.Reeves-Reed Arboretum

`Gemini-1.5-Pro:`
The golf club located at the intersection of Springfield and Houdaille Quarry is "Baltusrol Golf Club".
You can clearly see the label for Baltusrol Golf Club positioned right above Springfield Ave, with the golf flag
icon situated directly at the point where Springfield Ave intersects with Houdaille Quarry.

`GPT-4o-mini:`
The correct option is "2. Baltusrol Golf Club."
Explanation:
In the image, the location of Houdaille Quarry and Springfield can be found near the center at the junction of major
 roads. The Baltusrol Golf Club is marked nearby, indicating it is situated at that intersection. The other options,
 such as Brooklake Country Club, East Orange Golf Course, and Reeves-Reed Arboretum, do not appear to be located at
the intersection of Springfield and Houdaille Quarry.

`Calude-3.5-Sonnet:`
Based on the map image, the correct answer is: "2. Baltusrol Golf Club"
Explanation:
The map shows Baltusrol Golf Club located near the intersection of Springfield Avenue and Houdaille Quarry. You can
see "Baltusrol Golf Club" marked on the map just south of where Route 24 meets I-78, which is very close to where
Springfield Avenue and Houdaille Quarry (both visible on the map) would intersect. Therefore, Baltusrol Golf Club is
 the golf club located at the intersection of Springfield Avenue and Houdaille Quarry.

`Llama-3-VILA1.5-8B:`
The answer is 3.East Orange Golf Course. The location of East Orange Golf Course is at the intersection of
Springfield and Houdaille Quarry, as shown in the image. The image is a screenshot of a map, and the location of
East Orange Golf Course is marked with a red pin.

*Listing 10.* Example evaluation of MapEval-Visual `Green` : Correct Answer. `Red` : Wrong Answer.

`Context:`

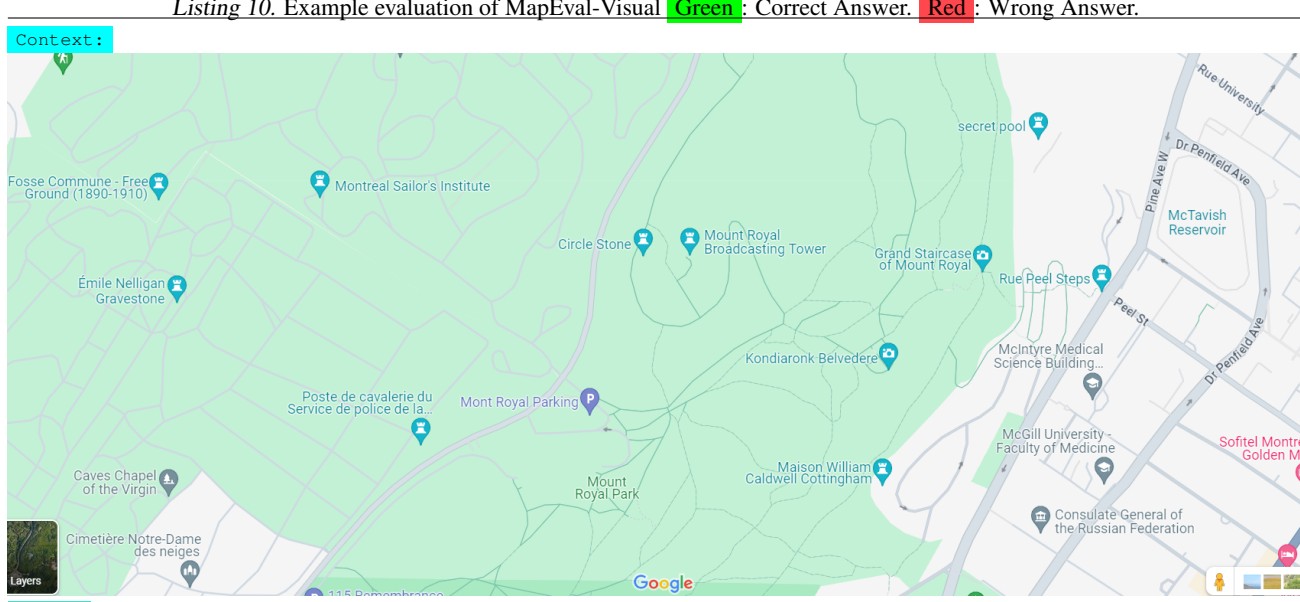

`Query:`

I am presently visiting  Mount Royal Park . Could you please inform me about the nearby historical landmark?

`Prompt:`

Choose the correct option from the image also show the answer with visible quotation and give proper explanation.

`Options:`

`1.` Circle Stone
2.secret pool
3.Maison William Caldwell Cottingham
4.Poste de cavalerie du Service de police de la Ville de Montreal

`Gemini-1.5-Pro:`
The correct answer is "Maison William Caldwell Cottingham".
The map shows the location of "Mount Royal Park". Just to the east of that marker is the "Maison William Caldwell Cottingham". The other options are located further away from Mount Royal Park.

`GPT-4o-mini:`
The correct option is:"3. Maison William Caldwell Cottingham"
Explanation:
Maison William Caldwell Cottingham is a historical landmark located near Mount Royal Park. It reflects the architectural heritage and history of the area, providing insights into the past. The other options, while interesting, do not hold the same historical significance as Maison William Caldwell Cottingham.

`Calude-3.5-Sonnet:`
Based on the map image provided, the correct answer is:"Circle Stone"
Explanation:
The question asks about a nearby historical landmark to Mount Royal Park. Looking at the map, we can see Mount Royal Park clearly labeled. Among the given options, "Circle Stone" is the closest historical landmark visible on the map near Mount Royal Park. It's marked with a pin icon just north of the park area.
The other options are either farther away or not visible on this particular map view:
– "secret pool" is visible but seems to be farther to the northeast.
– "Maison William Caldwell Cottingham" is visible but located to the southeast of the park.
– "Poste de cavalerie du Service de police de la Ville de Montreal" is visible but located to the west of the park.

`Llama-3-VILA1.5-8B:`
The answer is 3.Maison William Caldwell Cottingham.
Maison William Caldwell Cottingham is a historic house museum located in Montreal, Quebec, Canada. It is located in the Mount Royal Park, which is a popular tourist attraction in Montreal. The house was built in 1851 and is a fine example of Gothic Revival architecture. It is now a museum that showcases the history and culture of Montreal..

*Listing 11.* Example evaluation of MapEval-Visual  Green : Correct Answer.  Red : Wrong Answer.

Context:

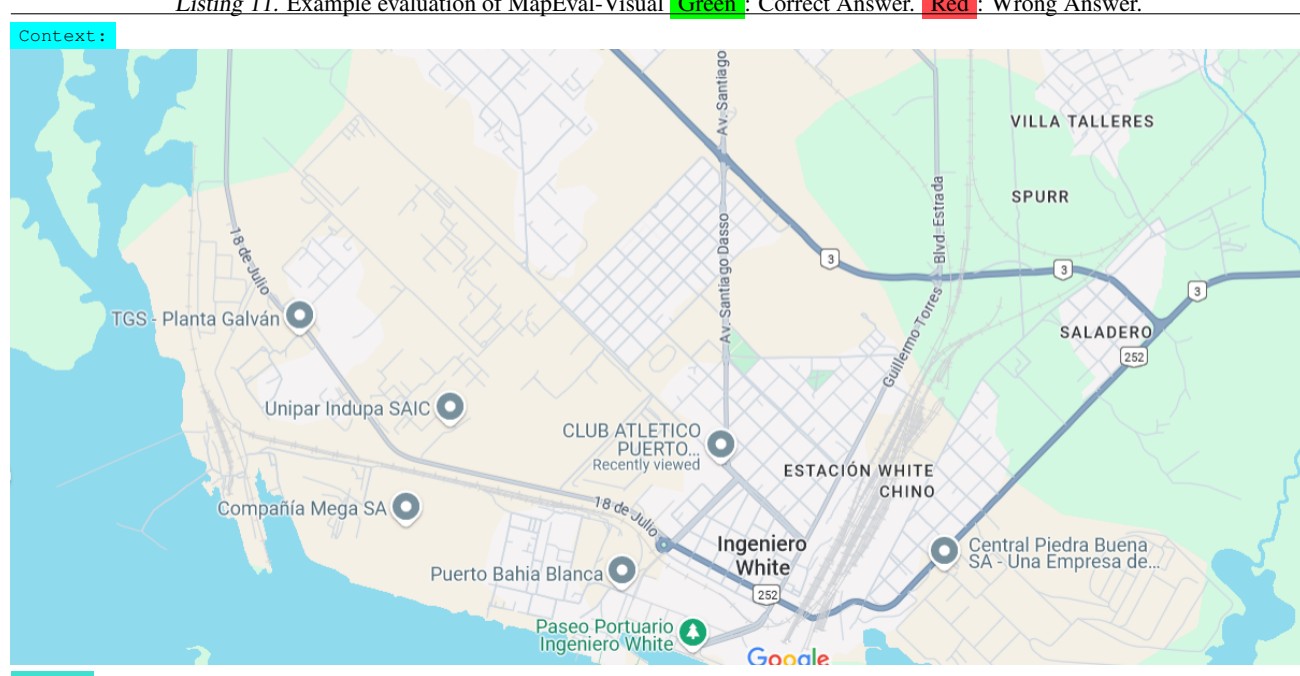

Query:
Which school is closed to Club Atletico Puerto Comercial?

─────

Prompt:
Choose the correct option from the image also show the answer with visible quotation and give proper explanation.

─────

Options:
0.  Unanswerable
1. Secondary school No. 25
2. PROVINCIAL SCHOOL No. 67 TOMAS ESPORA
3. Province of Buenos Aires SCHOOL No. 507
4. SCHOOL No.5 Province of Buenos Aires

─────

Gemini-1.5-Pro:
The answer is "0) Unanswerable".
The map provided does not show any schools. Therefore, it is impossible to determine which school is closed to Club
Atletico Puerto Comercial based solely on the provided image.

GPT-4o-mini:
To determine which school is closest to Club Atletico Puerto Comercial, we need to analyze the locations of the
listed schools in relation to the club's position on the map.
Analysis:
- Club Atletico Puerto Comercial is located near the center bottom of the map.
- The other schools must be found or deduced based on their general proximity to this location
From the visible landmarks and overall arrangement on the map, PROVINCIAL SCHOOL No. 67 TOMAS ESPORA appears to be
the closest to Club Atletico Puerto Comercial.

Calude-3.5-Sonnet:
Based on the information provided in the image, the correct answer is:0) Unanswerable
Explanation:
The map does not show any schools or educational institutions near Club Atletico Puerto. While it shows various
landmarks and businesses, including "Club Atletico Puerto", there is no information about any schools visible on
this map. The options provided mention specific schools, but none of these are shown or labeled on the map image.
Given the limited information in the image and the absence of any school locations, the question cannot be answered
based solely on what's shown here, making "Unanswerable" the most appropriate choice.

Llama-3-VILA1.5-8B:
The answer is 1) Secondary school No. 25.
The image shows a map of Puerto Comercial, Argentina, with various locations marked. One of the locations marked is
a secondary school, which is the closest school to Club Atletico Puerto Comercial. The answer is 1) Secondary school
 No. 25.

*Listing 12.* Example evaluation of MapEval-Visual Green : Correct Answer. Red : Wrong Answer.

Context:

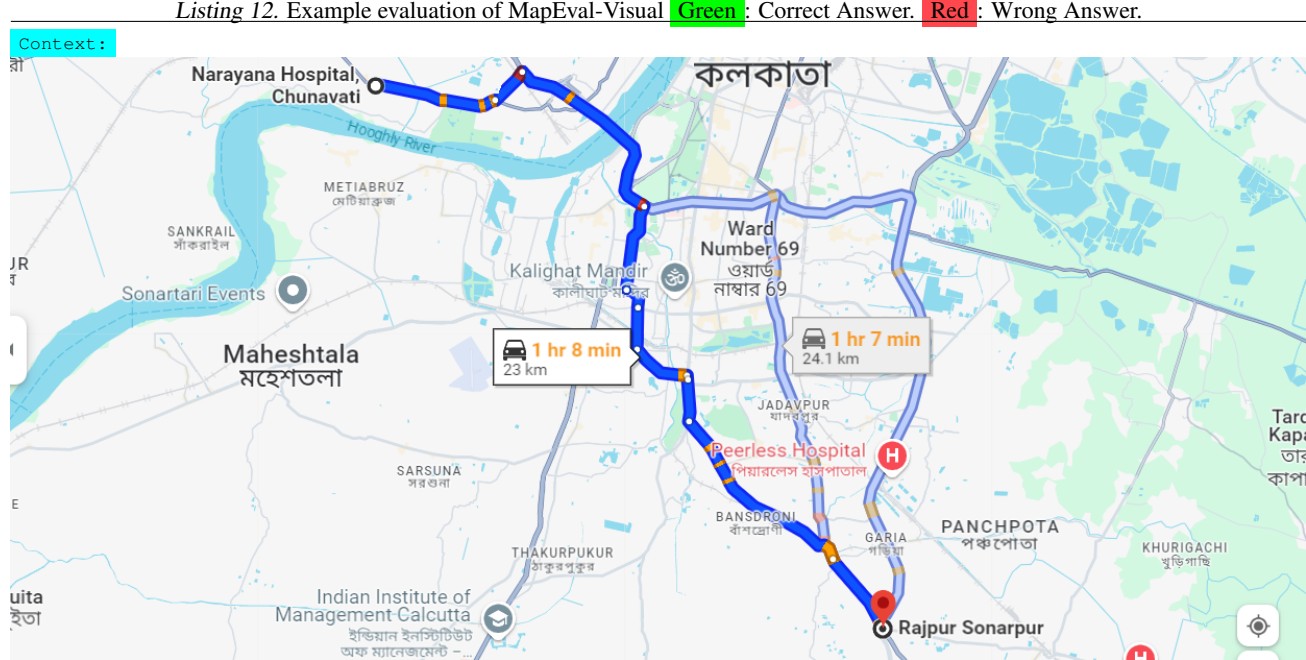

Query:
How far is Rajpur Sonarpur in West Bengal, India, from Narayana Hospital, Chunavati, Andul Rd, Howrah, West Bengal,
if I drive?

Prompt:
Choose the correct option from the image also show the answer with visible quotation and give proper explanation.

Options:
1.  23 km
2.  23.2km
3.  24.1km
4.  27.6km

Gemini-1.5-Pro:
The image clearly shows the total distance as "23 km".
Therefore, the correct answer is "1) 23 km".

GPT-4o-mini:
Based on the provided information and the image, the distance from Narayana Hospital, Chunavati, Howrah, to Rajpur
Sonarpur is approximately 23 km.
The answer is option 1) "23 km"
Explanation:
- The route consists of several parts, with the total distance calculated and displayed in the map:
- The total time shown on the map indicates a travel duration of around 1 hour and 8 minutes for the distance of 23
km.

Calude-3.5-Sonnet:
Based on the information provided in the image, the correct answer is:"23 km"
Explanation:
The map shows a route from Narayana Hospital, Chunavati (which is near the Hooghly River in the northwest of the map
) to Rajpur Sonarpur (in the southeast of the map). The route is highlighted in blue. In the bottom left corner of
this blue route, there's a distance and time estimate clearly visible:"1 hr 8 min 23 km"

Llama-3-VILA1.5-8B:
The answer is 27.6km.
The distance from Narayana Hospital, Chunavati, Andul Rd, Howrah, West Bengal, to Rajpur Sonarpur is 27.6 kilometers
. The route you described involves traveling along Andul Rd, NH 12, Alipore Rd, Garia Main Rd/Netaji Subhash Chandra
 Bose Rd, and passing by Jharna Ghee And Honey on the right. The total distance covered is 27.6 kilometers.

*Listing 13.* Example evaluation of MAPEVAL-VISUAL Green : Correct Answer. Red : Wrong Answer.

`Context:`

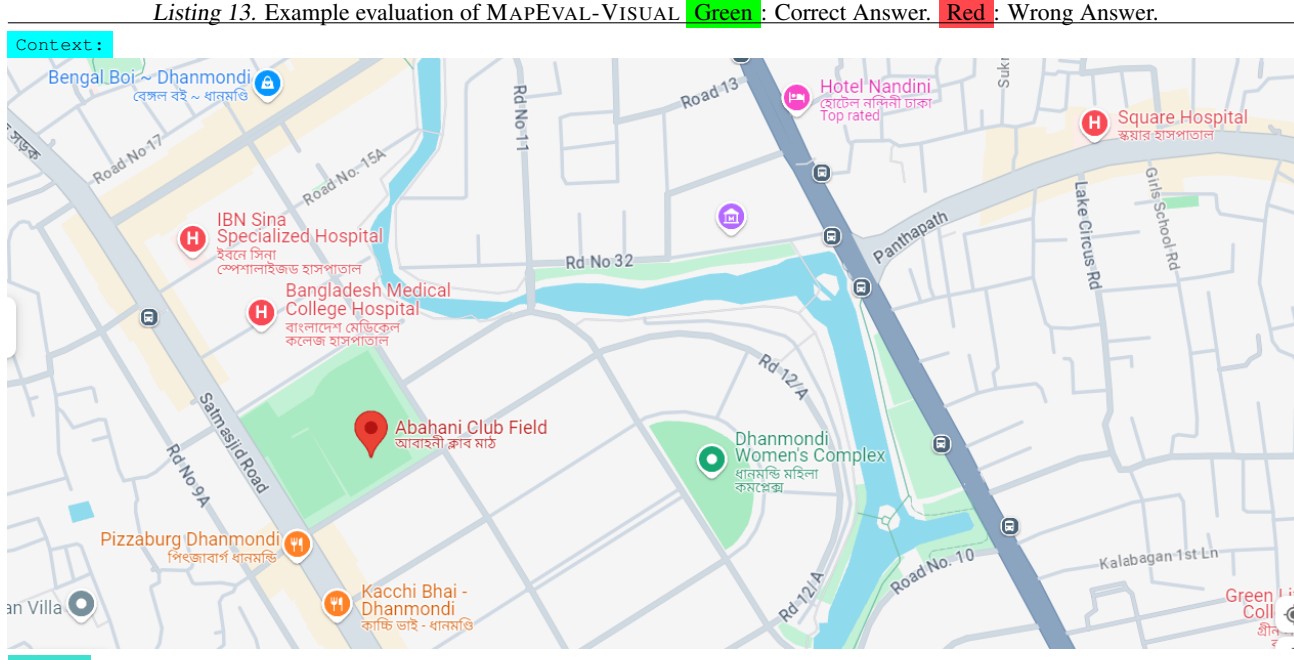

`Query:`

I'm staying at Hotel Nandini and heading to Dhanmondi Abahani Field to meet my friend for a game. I plan to check my blood pressure after playing. Can you suggest a suitable hospital for this?

`Prompt:`

Choose the correct option from the image also show the answer with visible quotation and give proper explanation.

`Options:`
1.Square Hospital
2.Kacchi Bhai – Dhanmondi
3.Bangladesh Medical College Hospital
4.IBN Sina Specialized Hospital

`Gemini-1.5-Pro:`
The best option is "1. Square Hospital". Here's why:
Familiarity: Staying with a healthcare provider you're familiar with is generally a good idea, especially for something like checking blood pressure.
Location: While not the absolute closest, Square Hospital is still relatively convenient to both Hotel Nandini and Abahani Field. It's a short trip away from both locations.

`GPT-4o-mini:`
The most suitable option for checking your blood pressure after playing at Dhanmondi Abahani Field is: "IBN Sina Specialized Hospital". Explanation:
1. Proximity: IBN Sina Specialized Hospital is located close to Abahani Club Field, making it convenient to access right after your game.
2. Alternatives: The other options may be farther away or may not be as specialized for such treatments.

`Calude-3.5-Sonnet:`
The most suitable option for checking your blood pressure after playing at Abahani Club Field would be: "Square Hospital". Explanation:
1. You mentioned that you usually get your treatment at Square Hospital, which is visible on the map in the upper right corner.
2. Square Hospital is relatively close to both Hotel Nandini (where you're staying) and Abahani Club Field (where you'll be playing).
3. While Bangladesh Medical College Hospital and IBN Sina Specialized Hospital are also visible on the map and are valid medical facilities, Square Hospital is your usual treatment location, making it the most suitable choice for continuity of care.

`Llama-3-VILA1.5-8B:`
The answer is 4. IBN Sina Specialized Hospital.
Explanation: The image shows a map of the area around Hotel Nandini and Dhanmondi Abahani Field. IBN Sina Specialized Hospital is located near Hotel Nandini and is a suitable option for checking blood pressure. It is also close to Dhanmondi Abahani Field.

*Listing 14.* Example evaluation of MAPEVAL-VISUAL  Green : Correct Answer.  Red : Wrong Answer.

`Context:`

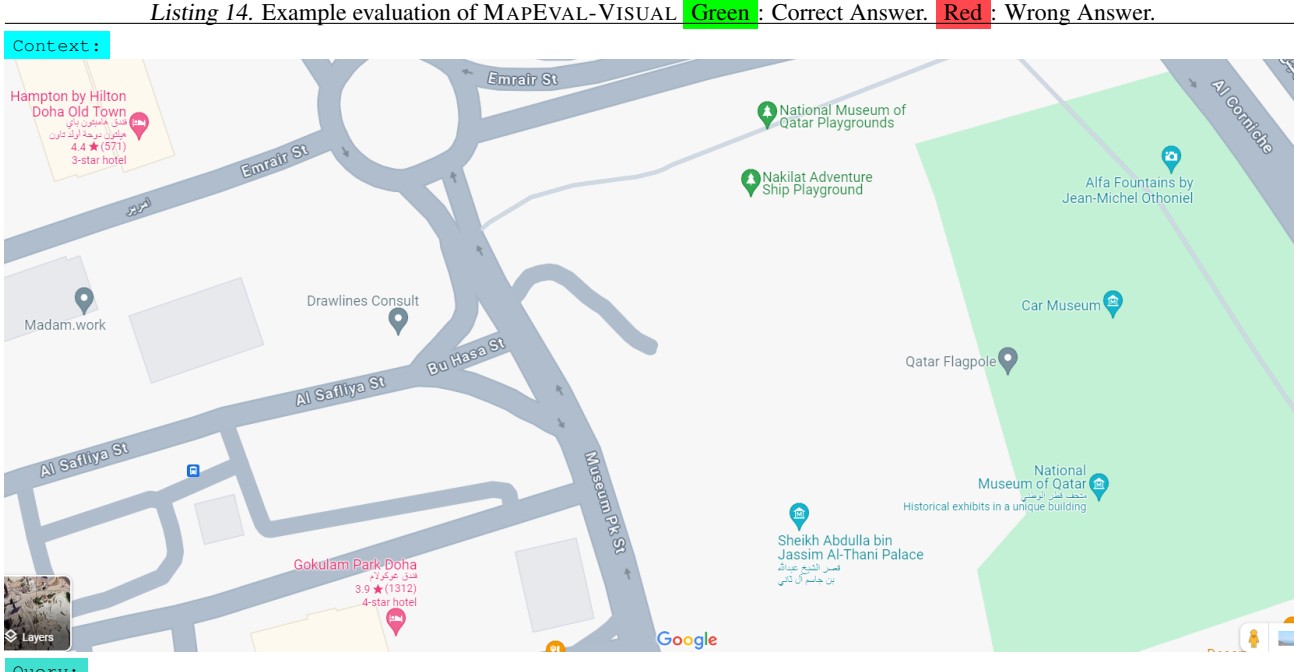

`Query:`

What is the nearest any Museum from Qatar Flagpole ?

───────

`Prompt:`

Choose the correct option from the image also show the answer with visible quotation and give proper explanation.

───────

`Options:`

1. Car Museum
2.Sheikh Abdulla bin Jassim Al-Thani Palace
3.National Museum of Qatar Playgrounds
4.National Museum of Qatar

───────

`Gemini-1.5-Pro:`

The correct option is "1) Car Museum".
From the image, you can see the "Qatar Flagpole" location marked. The **"Car Museum"** is located just above the Qatar Flagpole, making it the nearest museum.

`GPT-4o-mini:`

The nearest museum from the Qatar Flagpole is "National Museum of Qatar".
Explanation:
When examining the map, the National Museum of Qatar is directly adjacent to the Qatar Flagpole, while the other options are farther away. The Car Museum and Sheikh Abdulla bin Jassim Al-Thani Palace are located nearby, but the National Museum of Qatar is the closest, as indicated by its proximity on the map.

`Calude-3.5-Sonnet:`

Based on the map image provided, the correct answer is: "Car Museum"
Explanation:
The Qatar Flagpole is visible on the map, and among the museum options listed, the Car Museum appears to be the closest to it. The Car Museum icon is located just slightly northwest of the Qatar Flagpole marker on the map.
The National Museum of Qatar Playgrounds and the National Museum of Qatar are both further away to the north. Sheikh Abdulla bin Jassim Al-Thani Palace is located further south of the flagpole.

`Llama-3-VILA1.5-8B:`

The nearest museum from Qatar Flagpole is the National Museum of Qatar. The answer is option 4. The National Museum of Qatar is located on the Qatar Flagpole, as indicated on the map. The National Museum of Qatar is a museum in Doha, Qatar, dedicated to the history and culture of Qatar. It is the largest museum in the country and one of the largest in the Middle East. The museum is located on the Corniche, a waterfront promenade in Doha, and is easily accessible by public transportation. The museum features a variety of exhibits, including archaeological artifacts, historical documents, and contemporary art. The museum also offers educational programs and workshops for visitors of all ages.

