# OpenReview forum: "MapEval: A Map-Based Evaluation of Geo-Spatial Reasoning in Foundation Models"
_ICML.cc/2025/Conference — ICML 2025 spotlightposter_

### Official Review · Reviewer_V6Fj · 2025-02-26

**Overall Recommendation:** 5

**Summary:**

This paper presents a novel benchmark dataset called MapEval constructed based on Google Maps for various map-based geospatial reasoning question answering. The dataset consists of three components: MapEval-Textual, MapEval-Visual, and MapEval-API which correspond to different types of geospatial questions that involves text, map images, and API calls.

Strength:
1. The MapEval presents a very unique and important dataset for the whole AI community, especially for the geospatial artificial intelligence community in general. The dataset is very useful to evaluate LLMs' performance on different geospatial reasoning capabilities.
2. The paper conducts a systematic evaluation across 28 different LLMs and provides a very comprehensive view of the presented GeoQA challenge.
3. The limitations of the current LLMs are highlighted and explored in great details which facilitates the future LLM and Geospatial LLM development.

Suggestion:
1. Can you describe the way how you select the geographic questions? Many QA benchmark works start at collecting important question sets such as HotpotQA and Web Questions. Where do you collect these 700 questions? How do you define the overall question types?
2. Any reason why the performance of Claude-3-5-Sonnet (90%) can outperform human (65%) in such a large margin on the Unanswerable type in Table 5? The same can be seen in Table 3.
3. If I understand correctly, the authors use MapQaTor to cache all API call responses and create a static database for model evaluation. If the generated structured API calls are a little bit different from the golden API calls, then they will get no results from the cached database but they might be able to get the correct answers if using the real Google APIs. Will this systematically penalize the model performance?

**Claims And Evidence:**

Yes

**Essential References Not Discussed:**

No.

**Experimental Designs Or Analyses:**

See above

**Methods And Evaluation Criteria:**

Yes

**Other Comments Or Suggestions:**

See above.

**Other Strengths And Weaknesses:**

See above.

**Questions For Authors:**

See above

**Relation To Broader Scientific Literature:**

This paper has a significant contribution to the GeoAI community as well as the LLM research. It will also benefit the general public since this dataset can be used for evaluating LLMs for various geospatial tasks that are conducted daily by the general public.

**Theoretical Claims:**

No theoretical claim.

---

> ### Author Rebuttal · Authors · 2025-03-30
>
> Thank you for your recognition of the applicability, effectiveness of our proposed benchmarking task, and comprehensive experiments in our work!
> > Q1: Can you describe the way how you select the geographic questions? Many QA benchmark works start at collecting important question sets such as HotpotQA and Web Questions. Where do you collect these 700 questions? How do you define the overall question types?
>
> For defining question types, we conducted an extensive literature review on geospatial and geographic question answering [1-6]. However, since our benchmark differs from prior works by focusing specifically on map-based user queries, we established our own question types based on the kinds of questions users typically ask on map services. To ensure coverage and relevance, we manually annotated 700 questions, drawing from both the literature review and our own experiences with daily map usage.
> > Q2: Any reason why the performance of Claude-3-5-Sonnet (90%) can outperform human (65%) in such a large margin on the Unanswerable type in Table 5? The same can be seen in Table 3.
>
> Claude-3.5-Sonnet systematically marks a question as Unanswerable when the necessary information is not available in the provided context. In contrast, human participants tend to rely on intuition or external knowledge to select the most plausible option from the given choices, even when the correct answer is not explicitly present. This difference in approach leads to a significant performance gap, as the model strictly adheres to the available context while humans may introduce subjective reasoning.
> > Q3: If I understand correctly, the authors use MapQaTor to cache all API call responses and create a static database for model evaluation. If the generated structured API calls are a little bit different from the golden API calls, then they will get no results from the cached database but they might be able to get the correct answers if using the real Google APIs. Will this systematically penalize the model performance?
>
> In MapEval-API, agents can query details of specific places, list of nearby places around a location, route between places, travel duration and distance between places. Here you have a misunderstanding that agents generate structured API calls. Rather agents have access to simplified functions. For example, to get travel duration between places agents need to call TravelTimeTool(origin, destination, travelMode). This function then generates the actual structured API calls, which mitigates small variations in API calls. So, if an agent needs the driving duration from place A to B, it needs to call TravelTimeTool(placeId_A, placeid_B, ‘drive’). No other function call is valid in this scenario. In table 8 (in the main paper), you can see we have cached enough API calls in our database. So, the model performance is not penalized due to absence of cached data. This claim can be justified by comparing Claude-3.5-Sonnet’s performance in MapEval-Textual (66.33%) and MapEval-API (64%). If the model were penalized, the performance of MapEval-API would have been much lower.
>
> **References:**
>
> [1] Kefalidis, Sergios-Anestis, et al. "Benchmarking geospatial question answering engines using the dataset GeoQuestions1089." International semantic web conference. Cham: Springer Nature Switzerland, 2023.\
> [2] Hamzei, Ehsan, et al. "Place questions and human-generated answers: A data analysis approach." Geospatial Technologies for Local and Regional Development: Proceedings of the 22nd AGILE Conference on Geographic Information Science 22. Springer International Publishing, 2020.\
> [3] Mai, Gengchen, et al. "Geographic question answering: challenges, uniqueness, classification, and future directions." AGILE: GIScience series 2 (2021): 8.\
> [4] Punjani, Dharmen, et al. "Template-based question answering over linked geospatial data." Proceedings of the 12th workshop on geographic information retrieval. 2018.\
> [5] Chen, Wei, et al. "A synergistic framework for geographic question answering." 2013 IEEE seventh international conference on semantic computing. IEEE, 2013.\
> [6] Mai, Gengchen, et al. "On the opportunities and challenges of foundation models for geospatial artificial intelligence." arXiv preprint arXiv:2304.06798 (2023).

---

### Official Review · Reviewer_irQQ · 2025-03-06

**Overall Recommendation:** 3

**Summary:**

This paper introduces a geospatial benchmark called MapEval. It covers textual, visual and API-related tasks, and evaluates a set of close-source and open-source LLMs and VLMs. The results highlight the gap between close-source and open-source models and between current foundation models and humans, suggesting potential improvement for map-related abilities in foundation models.

**Claims And Evidence:**

The proposed benchmark covers textual, API and visual tasks and is close to real-world scenarios, and the gap for improvement is validated through extensive experimental results. The appendix provides detailed comparison regarding different models from many perspectives.

**Essential References Not Discussed:**

I suggest including more discussion on benchmarking and improving general 2D spatial reasoning (i.e. not specially for maps or geospatial tasks) in LLMs and VLMs, such as [1-4]. I wonder whether they have similar conclusions or can be helpful in the proposed geospatial benchmark.

[1] Yang, Jianwei, et al. "Set-of-mark prompting unleashes extraordinary visual grounding in gpt-4v."

[2] Ramakrishnan, Santhosh Kumar, et al. "Does Spatial Cognition Emerge in Frontier Models?."

[3] Tang, Yihong, et al. "Sparkle: Mastering basic spatial capabilities in vision language models elicits generalization to composite spatial reasoning."

[4] Li, Chengzu, et al. "Topviewrs: Vision-language models as top-view spatial reasoners."

**Experimental Designs Or Analyses:**

1. It is good to see large LLMs with 70 and 90B parameters are included in the experiments. However, for VLMs, current evaluation only considers models with less than 10B parameters. Larger VLMs (e.g. Qwen2.5-VL 72B, InternVL2.5-78B), since they have stronger OCR and reasoning abilities and are supposed to narrow the gap between close-source models.
2. I check the examples in the appendix and find some questions:

  - For Listing 1, models fail to answer correctly due to inability to convert geospatial coordinates to actual distances. It also requires math capabilities, as validated by the experiments with calculators. These abilities somewhat deviate from "map-based reasoning", which is the focus of the proposed benchmark.

  - For Listing 12, it seems that the intermediate points in the query do not appear on the map.

  - For Listing 13, if I understand correctly, the answer should be 19.4/5.4=3.59 hours, while the options are all in minutes. Besides, the visual context seems to be unnecessary to answer this question.

**Methods And Evaluation Criteria:**

1. The number of samples in the benchmark is somewhat insufficient, especially considering each single task. For instance, some models in Table 3 exhibit closer performance in "place info" with only 1-2% difference. Considering there are only 64 samples in the textual place info task, the actual difference is only one or two samples. This data size is not sufficient to compare different models with statistical significance.
2. How to prevent LLMs/VLMs from using their pretrained knowledge (e.g. Knowledge of POIs in some cities) to answer the queries? The sentence "without using any external knowledge or assumptions" in the prompt may not necessarily be effective. Although the authors use LLMs to filter out questions that can be answered easily in Appendix B.2, this seems to manual process. I suggest providing a baseline with no textual/API/visual context to demonstrate that external context is necessary for answering these questions.
3. The questions are all designed as simple MCQs. While this is supposed to be an effective evaluation approach, open-ended questions or more complex responses (e.g. TravelPlanner) may be more suitable for certain tasks like trip planning.

**Other Comments Or Suggestions:**

Many figures and examples are provided in the appendix but frequently referred in the main paper, affecting the coherence and integrity of the paper. It is recommended that the discussion text related to the figures in the appendix should also be placed in the appendix, with only a brief mention in the main text.

**Other Strengths And Weaknesses:**

N/A

**Questions For Authors:**

1. The benchmark uses Google Map as the source. Can it be extended to other map services, or other modalities (e.g. Satellite images in Google Map)? Discussion is encouraged.

**Relation To Broader Scientific Literature:**

N/A

**Theoretical Claims:**

N/A

---

> ### Author Rebuttal · Authors · 2025-03-30
>
> > Q1: The benchmark uses Google Map as the source. Can it be extended to other map services, or other modalities (e.g. Satellite images in Google Map)?
>
> Yes, the benchmark can be extended to other map services. The latest MapQaTor update integrates OpenStreetMap, Mapbox, and TomTom, broadening applicability. While satellite images can be used, we focus on digital map snapshots as they better reflect everyday interactions like navigation and place searches.
>
> > Q2: How can we prevent LLMs/VLMs from using pretrained knowledge to answer queries? A baseline with no textual/API/visual context could demonstrate that external context is necessary.
>
> We evaluated the top-performing model Claude-3.5-Sonnet on 300 MCQs used in MapEval-Textual and MapEval-API without textual/API context. The overall accuracy is 6.67%, demonstrating the necessity of external context.
>
> > W1: The evaluation includes large LLMs, but only smaller VLMs (<10B parameters) are considered. Larger models (e.g. Qwen2.5-VL 72B, InternVL2.5-78B) might narrow the gap with closed-source models.
>
> MapEval is agnostic to the model families or sizes and any foundation models of corresponding modalities can be studied. Based on your suggestion, we performed additional experiments with larger Qwen2.5-VL-72B, and Llama3.2-90B-Vision, and the results (Table 1) confirm a reduced gap between closed (61.65%) and open-source (60.35%) models.
>
> |Model|Overall|POI|Nearby|Routing|Counting|Unanswerable|
> |-|:-:|:-:|:-:|:-:|:-:|:-:|
> |Qwen2.5-VL-72B|60.35|76.86|54.44|43.04|52.33|90.00|
> |Llama3.2-90B-Vision|50.38|73.55|46.67|41.25|36.36|25.00|
>
> *Table 1: Performance of larger VLMs in MapEval-Visual.*
> > W2: For Listing 1, models fail to correctly convert geospatial coordinates to distances and require math capabilities, as shown in the calculator experiments. These abilities slightly deviate from the "map-based reasoning" focus of the proposed benchmark.
>
> Numerical and mathematical capabilities are integral to map-based reasoning, as geospatial tasks inherently involve calculations such as distance estimation, travel time computation, and spatial relationships. Effective map-based reasoning requires models to first identify and understand different place types, recognize origin and destination locations, and interpret spatial context. Only then can they apply mathematical reasoning to solve tasks like route optimization, distance-based decisions, and travel cost estimations. Thus, rather than being separate from map-based reasoning, these numerical skills are essential for accurately processing and answering real-world geospatial queries.
> > W3: For Listing 12, it seems that the intermediate points in the query do not appear on the map.
>
> We have ensured that each question in our dataset can be answered using the given question text and the accompanying map snapshot. However, not all questions necessarily require the map snapshot for a correct response—some can be answered based on the textual information alone. Importantly, these types of queries constitute a small fraction of the dataset. These cases still align with real-world map usage scenarios where users may ask about locations or routes that are not fully displayed but can be inferred through reasoning.
> > W4: For Listing 13, the answer should be 19.4/5.4=3.59 hours, while the options are all in minutes. Besides, the visual context seems to be unnecessary to answer this question.
>
> We acknowledge the issue (options should be in hours, not minutes) and will correct this example in the final paper.
> > W5: The benchmark's sample size is limited, especially for tasks like "place info," where some models in Table 3 show only a 1-2% performance difference. With just 64 samples in the textual place info task, the actual difference is only one or two samples, making it insufficient for statistically significant model comparisons.
>
> It is true that some models in Table 3 show only a 1-2% difference in performance for the Place Info category. However, this is expected when benchmarking 19 LLMs, as minor variations naturally occur between models. That said, we carefully designed the benchmark to cover all practical variations within the 64 questions in this category. The Place Info task primarily focuses on factual attributes and spatial relationships, such as cardinal directions or straight-line distances, which inherently limit the number of distinct question types that can be meaningfully introduced. Simply increasing the number of questions would likely lead to redundancy rather than new insights into model performance. However, if the reviewers can suggest additional meaningful variations that we may have overlooked, we are open to expanding this category in the final version of the dataset.
> > W6: The questions are simple MCQs, which are effective for evaluation, but open-ended questions or more complex responses (e.g., TravelPlanner) may be better suited for tasks like trip planning.
>
> See W1 in Reviewer syVz’s rebuttal.

---

> > ### Comment · Reviewer_irQQ · 2025-04-02
> >
> > I have read the authors' response. I think "Simply increasing the number of questions would likely lead to redundancy rather than new insights into model performance" is a misunderstanding. More questions are necessary for statistical significance. Although the total number of samples may be sufficient, the number of samples in each task is far from enough. For the benchmarks mentioned in the response to reviewer Z5ay, despite only a few hundred samples in total, most of them do not report results and analysis on such many sub-tasks. I suggest either including more samples for each task or downplaying the analysis on each separate task.

---

> > > ### Author Response · Authors · 2025-04-02
> > >
> > > Thank you for your thoughtful feedback to improve our work!. We understand your concern about the statistical significance of per-task results due to the limited number of samples. As you suggested, we will clarify these limitations when discussing per task analyses in the camera-ready.
> > >
> > > Despite this, analyzing sub-task performance remains crucial for capturing model strengths and weaknesses. As shown in Table 3,4 and 5 in the paper, overall accuracy alone does not provide a complete picture.
> > >
> > > For instance, while Llama-3.2-90B scores 9% higher than Gemma-2.0-27B overall, the latter outperforms it by 5% in the *Nearby* category.
> > >
> > > | Model | Overall | Place Info | Nearby | Routing | Trip | Unanswerable |
> > > |----------|:----------:|:----------:|:----------:|:----------:|:----------:|:----------:|
> > > | Llama-3.2-90B | **58.33** | **68.75** | 66.27 | **66.67** | **38.81** | **30.00** |
> > > | Gemma-2.0-27B | 49.00 | 39.07 | **71.08** | 59.09 | 31.34 | 15.00 |
> > >
> > > *Table 1: Performance comparison of Llama-3.2-90B and Gemma-2.0-27B in MapEval-Textual task.*
> > >
> > > Similarly, models with similar overall scores, such as Claude-3.5-Sonnet and Gemini-1.5-Pro, exhibit notable differences in specific sub-tasks.
> > >
> > > | Model | Overall | Place Info | Nearby | Routing | Trip | Unanswerable |
> > > |----------|:----------:|:----------:|:----------:|:----------:|:----------:|:----------:|
> > > | Claude-3.5-Sonnet | **66.33** | **73.44** | 73.49 | **75.76** | **49.25** | 40.00 |
> > > | Gemini-1.5-Pro | **66.33** | 65.63 | **74.70** | 69.70 | 47.76 | **85.00** |
> > >
> > > *Table 2: Performance comparison of Claude-3.5-Sonnet and Gemini-1.5-Pro in MapEval-Textual task.*
> > >
> > > Moreover, even though overall accuracy in MapEval-API is less than MapEval-Textual for all models, performance of Claude-3.5-Sonnet in *Trip* category greatly improved.
> > >
> > > | Task| Overall | Place Info | Nearby | Routing | Trip | Unanswerable |
> > > |----------|:----------:|:----------:|:----------:|:----------:|:----------:|:----------:|
> > > | MapEval-Textual | **66.33** | **73.44** | **73.49** | **75.76** | 49.25 | 40.00 |
> > > | MapEval-API | 64.00 | 68.75 | 55.42 | 65.15 | **71.64** | **55.00** |
> > >
> > > *Table 3: Performance comparison of Claude-3.5-Sonnet in MapEval-Textual and MapEval-API task.*
> > >
> > > These types of insights were not possible without the analysis of each subtask.
> > >
> > > That said, we acknowledge the trade-off between statistical robustness and detailed task-level analyses in our paper. However, analysis on each separate task can be beneficial to gain insights about the strengths and weaknesses of different models while being mindful of the limitations that we will clarify in revision.

---

### Official Review · Reviewer_Z5ay · 2025-03-08

**Overall Recommendation:** 3

**Summary:**

The paper introduces MapEval, a benchmark designed to evaluate the geospatial reasoning capabilities of foundation models across textual, API-based, and visual tasks. It comprises 700 multiple-choice questions covering spatial relationships, navigation, travel planning, and map interactions across 180 cities and 54 countries. This paper evaluates various foundation models, including both closed models, including GPT-4o, Claude-3.5-Sonnet, and Gemini-1.5-Pro, and open-sourced models, revealing significant performance gaps. The results demonstrate critical weaknesses for existing LLMs in spatial inference, including difficulty in handling distances, directions, route planning, and location-specific reasoning. The paper highlights the need for better geospatial AI models that integrate improved reasoning and API interactions to bridge the gap between foundation models and real-world navigation applications.

**Claims And Evidence:**

The evaluation is comprehensive to test the effectiveness of LLMs.

**Essential References Not Discussed:**

The references generally contain the representative studies in this field.

**Experimental Designs Or Analyses:**

The experiments to judge the accuracy of the answer look reasonable to me.

**Methods And Evaluation Criteria:**

There are several concerns regarding this evaluation benchmark. First, while the benchmark introduces a novel evaluation approach, the number of questions remains relatively small in scale. Second, the three task types could see significant performance improvements when integrated with appropriate tools. The difficulty of these questions seem to be limited. Third, some questions, as illustrated in the appendix, appear to be relatively superficial—for example, identifying travel time for a given route. Fourth, while certain LLMs do not perform well on this benchmark, it remains unclear how fine-tuning on such a small dataset would impact their performance on these tasks.

**Other Comments Or Suggestions:**

None

**Other Strengths And Weaknesses:**

Strengths:

S1. The paper introduces MapEval, a well-structured benchmark that evaluates geospatial reasoning across textual, API-based, and visual tasks, covering 180 cities and 54 countries. The benchmark includes 700 multiple-choice questions that span a range of real-world map interactions, such as navigation, travel planning, and spatial relationships.

S2. The study highlights significant gaps in the geospatial reasoning abilities of both closed and open-source models, showing their struggles with certain types of tasks.

Weaknesses:

W1. Despite the proposed benchmark evaluation dataset, its size (700 questions) remains relatively small compared to other LLM benchmark evaluation dataset.

W2. The paper does not explore how fine-tuning models specifically on MapEval would impact performance. Simple fine-tuning on part of the instances may greatly enhance the model performance. If that’s the case, the capability of answering these questions wouldn’t be an issue for LLM.

W3. Some questions, as observed in the appendix, appear to be relatively simple, such as estimating travel time, which may not fully test deep geospatial reasoning.

**Questions For Authors:**

Please response to the comments in W1-W3.

**Relation To Broader Scientific Literature:**

The paper propose to establish a benchmark evaluation dataset for assessing the performance of geospatial reasoning capabilities for LLMs. Such a benchmark dataset is not available in previous studies.

**Theoretical Claims:**

No mathematical proofs are provided in the paper.

---

> ### Author Rebuttal · Authors · 2025-03-30
>
> Thank you for your recognition of the applicability, effectiveness of our proposed benchmarking task, and comprehensive experiments in our work!
>
> > W1: Despite the proposed benchmark evaluation dataset, its size (700 questions) remains relatively small compared to other LLM benchmark evaluation dataset.
>
> Due to the high cost of both foundation models as well as tools/APIs, recent language models often tend to evaluate on a small number of sub-sampled datasets. For example ReACT [1] uses only 500 random samples from AlfWorld dataset, similarly Reflexion [2] uses only 100 examples from HotpotQA. Therefore, many recently proposed tool oriented or intense reasoning benchmark datasets are found to be reasonable in size in order to be cost-effective: API-Bank: [3] (400 instances), Logical-reasoning benchmark LogiQA: [4] (641 examples), the most popular problem solving (code generation benchmarks) HumanEval: [5] (164 instances only), CodeContests [6] (156 problems), Tau-bench [7] (165 problems), OS World [8] (369 problems), App world [9] (750 problems), TravelPlanner [10] (1.2K problems). Consequently, we carefully construct our problem instances balanced in size and covering different challenges.
>
> > W2: The paper does not explore how fine-tuning models specifically on MapEval would impact performance. Simple fine-tuning on part of the instances may greatly enhance the model performance. If that’s the case, the capability of answering these questions wouldn’t be an issue for LLM.
>
> We conducted additional experiments to assess the impact of fine-tuning on the performance of smaller models using the MapEval-Textual dataset. Specifically, we have split the dataset of 300 MCQs into train (97 questions) and test (203 questions) set and fine-tuned a selection of models on the train set and evaluated their performance on the test set. However, the results, as shown in Table 1, reveal that fine-tuning on MapEval does not lead to significant performance improvements (<5%) and remains remarkably lower than large capable models such as GPT-4o, Claude-3.5-Sonnet or Gemini 1.5 Pro which we already included in the paper. Rather, we believe our evaluation benchmark MapEval will promote future developments of new geo-spatial model with sophisticated fine-tuning and other learning methods.
>
> These findings suggest that the challenges in MapEval are not merely due to a lack of training exposure but reflect deeper limitations in LLMs' geospatial reasoning.
>
>
> | Model			| Pretrained | Finetuned |
> |---------------|:-------------:|:--------------:|
> | Phi-3.5-mini 		|   39.90      |    34.48      |
> | Llama-3.2-3B 	|   34.98      |    35.96      |
> | Qwen-2.5-7B  	|   41.87      |    43.35      |
> | Llama-3.1-8B 	|   46.31      |    44.33      |
> | Gemma-2.0-9B 	|   46.80      |    51.23      |
>
>
> *Table 1: Performance Comparison of Open-Source Models on MapEval-Textual (Test set), before and after fine-tuning.*
>
> > W3: Some questions, as observed in the appendix, appear to be relatively simple, such as estimating travel time, which may not fully test deep geospatial reasoning.
>
> While some questions (e.g., travel time estimation) appear simple, our dataset prioritizes real-world map usage scenarios over exclusively testing deep geospatial reasoning. It includes practical queries (multi-stop routing, proximity-based decisions) and complex tasks (route optimization, accessibility constraints) requiring spatio-temporal reasoning. Simple examples in the appendix aim to clarify concepts, but the full dataset contains advanced reasoning challenges as well. As we discussed, even for the humanly simple cases, the advanced foundation models capable of doing more complex reasoning in other tasks fall significantly behind in MapEval. However, we will expand on these in the final version of the paper with more illustrated examples.
>
> **References:**
>
> [1] Yao, S., et al. "ReAct: Synergizing Reasoning and Acting in Language Models." ICLR 2023.\
> [2] Shinn, N., et al. "Reflexion: Language agents with verbal reinforcement learning." NeurIPS 2024.\
> [3] Li, M., et al. "API-Bank: A Benchmark for Tool-Augmented LLMs." EMNLP 2023.\
> [4] Liu, J., et al. "LogiQA: A challenge dataset for machine reading comprehension with logical reasoning." IJCAI 2021.\
> [5]Chen, M., et al. "Evaluating large language models trained on code." arXiv:2107.03374, 2021.\
> [6] Li, Y., et al. "Competition-level code generation with alphacode." Science 378.6624 (2022).\
> [7] Yao, S., et al. "tau-bench: A Benchmark for Tool-Agent-User Interaction." arXiv:2406.12045, 2024.\
> [8] Xie, T., et al. "Osworld: Benchmarking multimodal agents for open-ended tasks." arXiv:2404.07972, 2024.\
> [9] Trivedi, H., et al. "AppWorld: A Controllable World of Apps and People." ACL 2024.\
> [10] Xie, J., et al. "TravelPlanner: A Benchmark for Real-World Planning." ICML 2024.

---

> > ### Comment · Reviewer_Z5ay · 2025-04-02
> >
> > Thanks for your reply. These responses are reasonable to me. I will raise my score accordingly.

---

> > > ### Author Response · Authors · 2025-04-03
> > >
> > > Thank you so much for your kind consideration!

---

### Official Review · Reviewer_syVz · 2025-03-12

**Overall Recommendation:** 3

**Summary:**

The authors introduce a benchmark designed to assess the map-based reasoning capabilities of foundation models. This benchmark consists of 700 multiple-choice questions covering locations, including 180 cities and 54 countries across tasks such as processing spatial relationships, navigation, travel planning, etc. The study evaluates 28 foundation models and finds that there still exists a significant performance gap compared to human capabilities, especially in complex map-based reasoning tasks.

**Claims And Evidence:**

The major claims are supported by their evaluation using different foundation models and their benchmark.

**Essential References Not Discussed:**

NA

**Experimental Designs Or Analyses:**

The focus of this work is the proposal of a new benchmark to evaluate the foundational model's geo-spatial reasoning capacities.

**Methods And Evaluation Criteria:**

The proposed methods and evaluation criteria generally make sense for assessing geospatial reasoning in foundation models. The authors make the benchmark relevant to real-world applications (e.g., navigation) and compare across different models, locations and tasks. The authors provided an error analysis, which offers insight into failure modes.

**Other Comments Or Suggestions:**

Se questions.

**Other Strengths And Weaknesses:**

The authors introduced a novel and well-structured benchmark for an interesting problem. The authors evaluate a wide range of models, providing insights into current limitations, and also discuss how to address failures and improve the reasoning ability in the future. One potential limitation is that the benchmark focuses on multiple-choice questions, which may not fully capture the open-ended nature of real-world spatial reasoning tasks.

**Questions For Authors:**

Can authors provide a more detailed discussion/insight on why different models perform differently on the datasets?
Is there any geo-specialized foundation models that the authors consider evaluating and would potentially outperform the current ones?

**Relation To Broader Scientific Literature:**

This work complements existing benchmarks in natural language processing by focusing on this specific domain (geo-spatial reasoning).

**Theoretical Claims:**

This is an application and benchmark work.

---

> ### Author Rebuttal · Authors · 2025-03-30
>
> Thank you for your recognition of the applicability, effectiveness of our proposed benchmarking task, and comprehensive experiments in our work!
> > W1: One potential limitation is that the benchmark focuses on multiple-choice questions, which may not fully capture the open-ended nature of real-world spatial reasoning tasks.
>
> In Section 3.1 (line 145 onwards) we discuss the motivations for MCQ based evaluation choice over open-ended ones.
>
> Besides, our dataset is designed to be flexible: the MCQ format was chosen for evaluation purposes, but the questions themselves are structured so that open-ended evaluation can be done simply by removing the answer choices. To further support this, we conducted an experiment where we removed answer choices from MapEval-Textual and MapEval-Visual questions and evaluated open-ended responses from **Claude-3.5-Sonnet** against ground truth using **O3-mini**. The results in Table 1 and 2 show Claude-3.5-Sonnet’s performance in the open-ended setting, demonstrating that our queries remain valid for such evaluations.
>
> |Evaluation|Overall|POI|Nearby|Routing|Trip|Unanswerable|
> |-|:-:|:-:|:-:|:-:|:-:|:-:|
> |Open-ended|55.33|43.75|43.37|69.70|67.16|55.00|
> |MCQ|66.33|73.44|73.49|75.76|49.25|40.00|
>
> *Table 1: Performance of Claude-3.5-Sonnet in MapEval-Textual*
>
> |Evaluation|Overall|PlaceInfo|Nearby|Routing|Counting|Unanswerable|
> |-|:-:|:-:|:-:|:-:|:-:|:-:|
> |Open-ended|51.88|67.7|51.11|35.00|43.18|65.00|
> |MCQ|61.65|82.64|55.56|45.00|47.73|90.00|
>
> *Table 2: Performance of Claude-3.5-Sonnet in MapEval-Visual*
>
> As discussed in Section 3.1 that while open-ended assessments are more natural they introduces additional challenges, particularly in automated grading. For example, O3-mini initially assessed Claude-3.5-Sonnet’s accuracy in the "Unanswerable" category as **55%** (Table 1), but manual inspection revealed an actual accuracy of **80%**--showing the limitations in an automated evaluation for open-ended setting.
>
> Additionally, open-ended evaluation doubles API costs since it requires two calls per query—one for generating responses and another for evaluation—unlike MCQ, which allows direct comparison. This makes large-scale assessments far more expensive. Moreover, as LLM API costs scale with token usage, long-form responses further amplify expenses.
>
> Thus, while open-ended evaluation is possible with MapEval if intended, our MCQ-based approach remains the more cost-effective and reliable method for benchmarking.
>
> > Q1: Can authors provide a more detailed discussion/insight on why different models perform differently on the datasets?
>
> In this work, we provide a detailed evaluation of various models on our datasets, highlighting their strengths and weaknesses in different categories. Our results show clear differences in model performance across fine-grained categories, with some models excelling in certain tasks and others facing challenges in specific areas. These insights, particularly discussed in Section 4.3 and Section 5, suggest that models may be better trained for certain reasoning aspects while struggling with others, which could be due to factors such as the nature of their training data or the types of tasks they were explicitly trained to handle.
>
> However, a more thorough causal analysis is challenging, as the full training procedures and datasets for these open-source foundation models are rarely disclosed. This lack of transparency limits our ability to directly analyze the root causes of the performance differences we observe. While we acknowledge this limitation, we believe that conducting such causal analysis falls outside the scope of this paper and will be explored in future work.
>
> > Q2: Is there any geo-specialized foundation models that the authors consider evaluating and would potentially outperform the current ones?
>
> At the time of our evaluation, we did not find any geo-specialized foundation models that met the specific requirements of our task. Most existing models in this domain are Vision-Language Foundation Models designed primarily for remote sensing images, which are not directly applicable to our evaluation. We have also addressed this in Appendix A.3.
>
> However, we evaluated **K2** [1], a model specifically designed for geoscience-related tasks and built on LLaMA-7B. However, as shown in Table 3, its performance on our benchmark was extremely poor, achieving an overall accuracy of only 20.33%, which is close to random guessing. Given its limited effectiveness across all categories, we decided not to include it in our main evaluations.
>
> |Overall|Place Info|Nearby|Routing|Trip|Unanswerable|
> |:-:|:-:|:-:|:-:|:-:|:-:|
> |20.33|25.00|20.48|15.15|20.90|20.00|
>
> *Table 3: Performance of K2 in MapEval-Textual*
>
> Nonetheless, if you refer us any specific model, we will report that.
>
> **References:**
>
> [1] Deng, C., et al. "K2: A foundation language model for geoscience." WSDM (2024).

---

### Decision · Program_Chairs · 2025-05-01

**Decision:**

Accept (spotlight poster)

**Comment:**

This paper presents MapEval, a novel and comprehensive benchmark for evaluating geospatial reasoning in foundation models across textual, visual, and API-based tasks. The benchmark is well-structured, covers a diverse set of cities and countries, and reveals critical limitations of current models, making a strong contribution to both the LLM and geospatial AI communities. While some reviewers noted limitations in benchmark scale and question complexity, the overall consensus highlights the benchmark's originality, relevance, and potential for future impact. I believe it is a clear accept.